# Top-down acetylcholine signaling via olfactory bulb vasopressin cells contributes to social discrimination in rats

Hajime Suyama [1], Veronica Egger [1] & Michael Lukas [1✉]

Social discrimination in rats requires activation of the intrinsic bulbar vasopressin system, but it is unclear how this system comes into operation, as olfactory nerve stimulation primarily inhibits bulbar vasopressin cells (VPCs). Here we show that stimulation with a conspecific can activate bulbar VPCs, indicating that VPC activation depends on more than olfactory cues during social interaction. A series of in vitro electrophysiology, pharmacology and immuno-histochemistry experiments implies that acetylcholine, probably originating from centrifugal projections, can enable olfactory nerve-evoked action potentials in VPCs. Finally, cholinergic activation of the vasopressin system contributes to vasopressin-dependent social dis-crimination, since recognition of a known rat was blocked by bulbar infusion of the muscarinic acetylcholine receptor antagonist atropine and rescued by additional bulbar application of vasopressin. Thus, our results implicate that top-down cholinergic modulation of bulbar VPC activity is involved in social discrimination in rats.

[1] Institute of Zoology, Neurophysiology, University of Regensburg, Regensburg, Germany. ✉email: michael.lukas@ur.de

Many mammals use olfactory cues as a fundamental communication tool, for the recognition and discrimination of individual conspecifics. A prominent example for a behavioral reaction that depends on olfaction-based discrimination of individuals is that ewes recognize the body odor of their own offspring and as a result, deny strange lambs to suckle[1]. Moreover, in prairie voles, the olfaction-based recognition of their mating partners enables them to establish monogamous pairbonds[2]. Rats and mice, the most common mammalian laboratory animals, also discriminate individual conspecifics via their odor signatures. This discrimination can then lead to various essential behavioral reactions[3]. To quantify the ability of rats to recognize individuals, so-called social discrimination tests are used[4]. Briefly, rats are exposed to a conspecific (sampling phase). After a short time of separation, rats are exposed to both, the same and a novel conspecific (discrimination phase). If the rats recognize the known conspecific, they investigate it less compared to the novel conspecific[4].

The peptidergic neuromodulator vasopressin (VP), an important mediator of various social behaviors in the mammalian brain[5], is a major player in facilitating social discrimination. For example, microinjection of VP into the olfactory bulb (OB) enhances social discrimination[6]. Furthermore, Tobin et al.[7] demonstrated the existence of an intrinsic bulbar VP system, consisting of VP-expressing cells (VPCs), and an impairment of social discrimination by the blockade of bulbar V1a receptors.

Recently, we classified these bulbar VPCs as non-bursting superficial tufted cells, featuring an apical dendritic tuft within a glomerulus, lateral dendrites along the top part of the external plexiform layer (EPL) and extended axonal ramifications, mostly within the entire EPL[8]. The dense apical tuft implies that VPCs receive excitatory inputs from the olfactory nerve (ON) just like other bulbar cells with glomerular tufts such as mitral cells (MCs)[9], e.g., during sampling of a conspecific's body odors. Intriguingly, we found that electric ON stimulation elicits primarily inhibitory postsynaptic potentials (IPSPs) as recorded at the soma. This dominant GABAergic input masks a delayed barrage of excitatory postsynaptic potentials (EPSPs). Therefore, if ON activation inhibits VPCs and thus cannot trigger VP release, it is unclear how intrinsic VP neurotransmission is at all possible during social discrimination. We hypothesize that to fully excite VPCs and allow VP release, non-ON inputs are required that either inhibit the GABAergic origin of the ON-evoked inhibition and/or deliver enough direct excitation to VPCs to outweigh the inhibition.

During social interaction, rats sample not only olfactory cues including urinary cues, but also other sensory cues, like vocalizations or touch[10]. Also, the sheer presence of a novel stimulus in their known environment may contribute to changes in their physiological and behavioral state. Thus, we suggest that states of arousal, attention, and/or the perception of non-olfactory cues during social interaction may be required for olfactory activation of bulbar VPCs. For the integration of such additional inputs into bulbar neurotransmission, top-down modulation is the most likely option. The most prominent modulatory centrifugal inputs to the OB are noradrenergic, serotonergic, and cholinergic fibers[11–13]. Those neuromodulatory systems were shown to be involved in changes of internal states[14–16], but more interestingly, they are also facilitators of social discrimination behavior, acting either directly in the OB or in other brain regions[17–19]. Thus, we propose them as candidate activators of the bulbar VP system.

This notion leads to the following questions: (1) Are bulbar VPCs indeed activated during social interactions (although they are predominantly inhibited by activated olfactory afferents)? (2) Can any of the candidate centrifugal neuromodulators increase excitation of bulbar VPCs? (3) If so, is centrifugal modulation of bulbar VPCs involved in VP-dependent social discrimination? We approach these questions by investigating neuronal activity and synaptic mechanisms in vitro as well as applying behavioral pharmacology in vivo.

## Results

**Social interaction activates bulbar VPCs.** As we recently demonstrated in vitro[8] that ON stimulation alone primarily inhibits VPCs, we infer that bottom-up olfactory nerve input is unlikely to excite VPCs. On the other hand, during social discrimination VPCs should be strongly excited as VP neurotransmission was shown to be essential for this behavior[7]. To test this hypothesis, male juvenile VP-eGFP rats[20] were exposed to either, water (control) or one of two stimuli that are commonly used in social discrimination paradigms, i.e., rat urine or a novel juvenile rat[4]. We then compared the neuronal activity of bulbar VPCs between these different types of stimulation as reflected in phosphorylated extracellular signal-regulated kinase (pERK) induction in eGFP-labeled VPCs (Fig. 1a). The average section size of the main OB (MOB, water $4.48 \pm 0.4$ mm$^2$, urine $4.58 \pm 0.3$ mm$^2$, rat $4.85 \pm 0.3$ mm$^2$) and average number of eGFP$^+$ VPCs per section (water $81 \pm 5$ VPCs, urine $79 \pm 5$ VPCs, rat $85 \pm 3$ VPCs) were not different across animal groups (section size; $p = 0.723$, ANOVA. eGFP$^+$ VPCs, Supplementary Fig. 1a; $p = 0.670$, ANOVA, water $n = 10$ rats, urine $n = 9$, rat $n = 10$)$_a$. While there is a substantial background of pERK$^+$ VPCs in vivo (see discussion), indeed, the exposure to a rat resulted in significantly higher fractions of pERK$^+$ VPCs than the exposure to water. However, rat exposure did not differ significantly from urine exposure (Fig. 1c + d; $p = 0.038$, ANOVA, rat vs. water $p = 0.011$, rat vs. urine $p = 0.197$, water vs. urine $p = 0.195$)$_b$. These findings indicate (1) that bulbar VPCs are indeed excitable in vivo and (2) that during social interaction additional yet unknown factors enable increased activation of VPCs.

The number of pERK$^+$ MCs was increased following the exposure to a rat compared to water or urine (Fig. 1b + e; $p = 0.042$, ANOVA, rat vs. water $p = 0.025$, rat vs. urine $p = 0.034$, water vs. urine $p = 0.949$)$_c$.

In addition, we analyzed activation of VPCs and mitral/tufted cells (M/TCs) in the accessory olfactory bulb (AOB), as the MOB and the AOB are both related to social discrimination in rats[21]. The average number of eGFP$^+$ VPCs per section was not different across animal groups (Supplementary Fig. 1c; $p = 0.539$, ANOVA, $n = 9$ rats per group)$_d$. However, pERK$^+$ VPCs were rarely observed (Supplementary Fig. 1d; water $2.1 \pm 0.7$ % ($0.4 \pm 0.1$ VPCs), urine $2.4 \pm 0.8$ % ($0.5 \pm 0.1$ VPCs), rat $3.4 \pm 0.8$ % ($1.0 \pm 0.2$ VPCs)) and the fraction of pERK$^+$ VPCs was similar between the different groups (Supplementary Fig. 1d; $p = 0.485$, ANOVA, $n = 9$ rats per group)$_e$. Thus, AOB VPCs are unlikely to play an important role in facilitating social discrimination compared to MOB VPCs. The number of activated AOB M/TCs was not different between groups (Supplementary Fig. 1f; $p = 0.164$, ANOVA)$_f$.

Taken together, an increased activation of VPCs was found following social interaction, and MOB VPCs showed more prominent activation than AOB VPCs. Thus, social interaction is likely to involve additional inputs (beyond ON input) that enable/unlock VPC activation via modulation.

**Cholinergic modulation triggers excitatory responses and action potentials during ON stimulation of VPCs.** The OB, especially the glomerular layer (GL) and the EPL where VPCs are located, receives many centrifugal projections that release neuromodulators. These projections originate from various brain areas[11–13]. The receptors of these neuromodulators are expressed

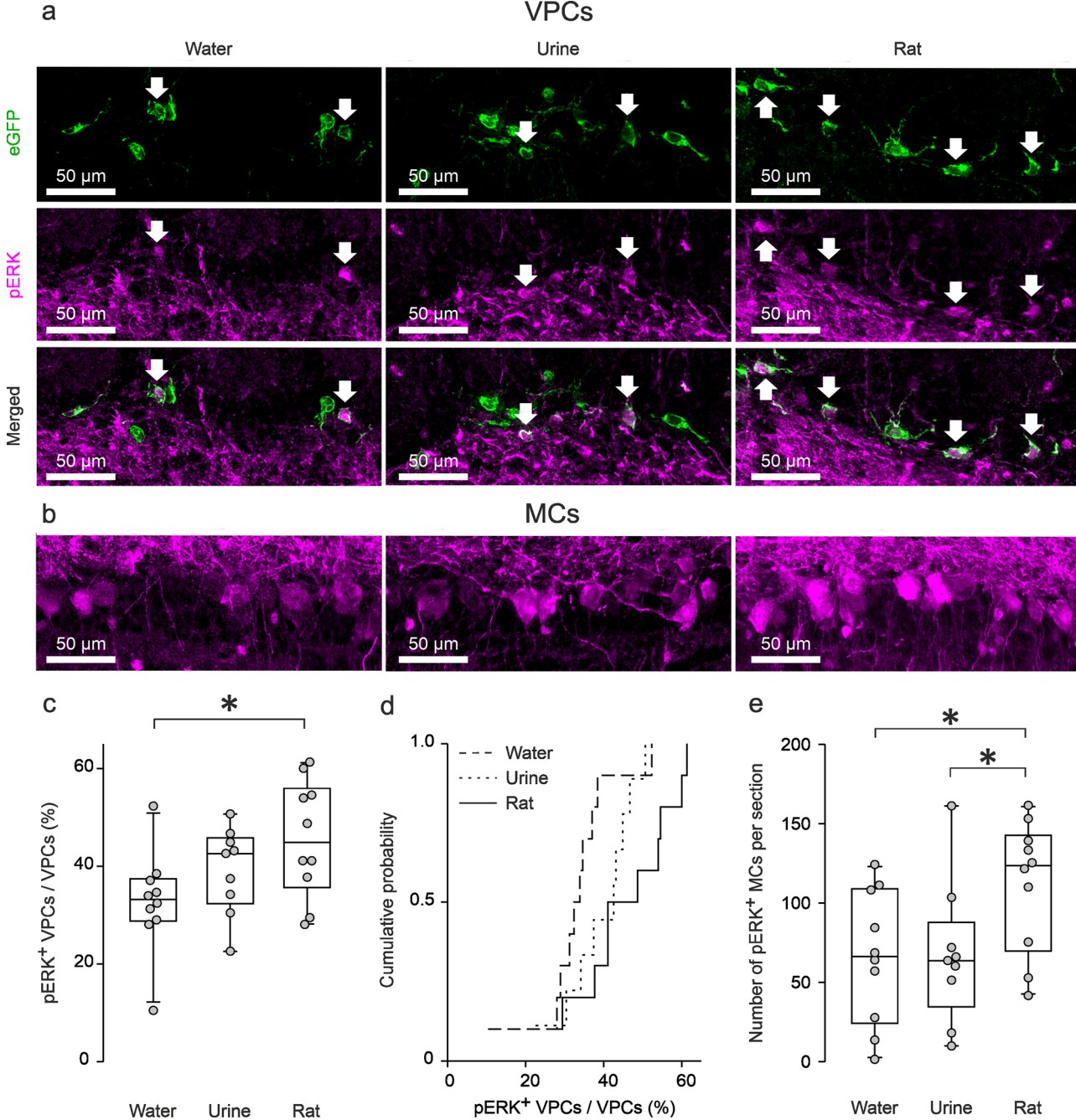

**Fig. 1 Social interaction activates bulbar VPCs.** Representative average z-projections of the olfactory bulb that were immune-stained for eGFP (green, CF488) and pERK (magenta, CF 594). Scale bar, 50 μm valid for all images in the same panel. **a** VPCs following water, urine, or rat stimulation. Arrows indicate cells that are double-labeled for eGFP and pERK. **b** pERK+ MCs. **c** Averaged fraction of pERK+ VPCs of all VPCs in different stimulation groups (%). One-way ANOVA, LSD for single comparison, *$p < 0.05$ rat vs. water. **d** Cumulative probability of averaged fraction pERK+ VPCs of all VPCs in different stimulation groups (%). **e** Averaged number of pERK+ MCs per section in different stimulation groups. One-way ANOVA, LSD for single comparison, *$p < 0.05$ rat vs. water and rat vs. urine. Data are presented as box-plots including first, median, and third quartiles with whiskers representing the range of data points and distribution of single data points. $n = 10$ rats (water), $n = 9$ rats (urine), $n = 10$ rats (rat).

in the entire MOB[22,23]. We hypothesized that such centrifugal neuromodulatory inputs to the MOB could mediate the increased VPC activity that we observed during social interaction. Thus, we performed whole-cell patch-clamp recordings from VPCs and stimulated the ON[8] while bath-applying neuromodulators (Fig. 2a).

Serotonin (5-HT, 20 μM) decreased the amplitudes of ON-evoked IPSPs to $69.7 \pm 10.1$ % of control (Fig. 2b + c; $p = 0.012$, $n = 8$ from 8 rats)$_g$. Noradrenaline (NA, 20 μM) reduced IPSP

amplitudes to $62.9 \pm 9.5$ % of control (Fig. 2d + e; $p = 0.011$, $n = 9$ from 8 rats)$_h$. Thus, 5-HT and NA modulation can reduce ON-evoked inhibition of VPCs but are unlikely to enable activation of VPCs as observed during social interaction (Fig. 1).

In contrast, wash-in of acetylcholine (ACh, 100 μM) switched inhibitory to excitatory responses in the majority of VPCs: 65 % of VPCs fired action potentials (APs, 13 out of 20 cells from 19 rats), 10 % of VPCs responded with EPSPs to ON stimulation. The rest all showed decreased inhibition (Fig. 2f + g; $p < 0.001$ for

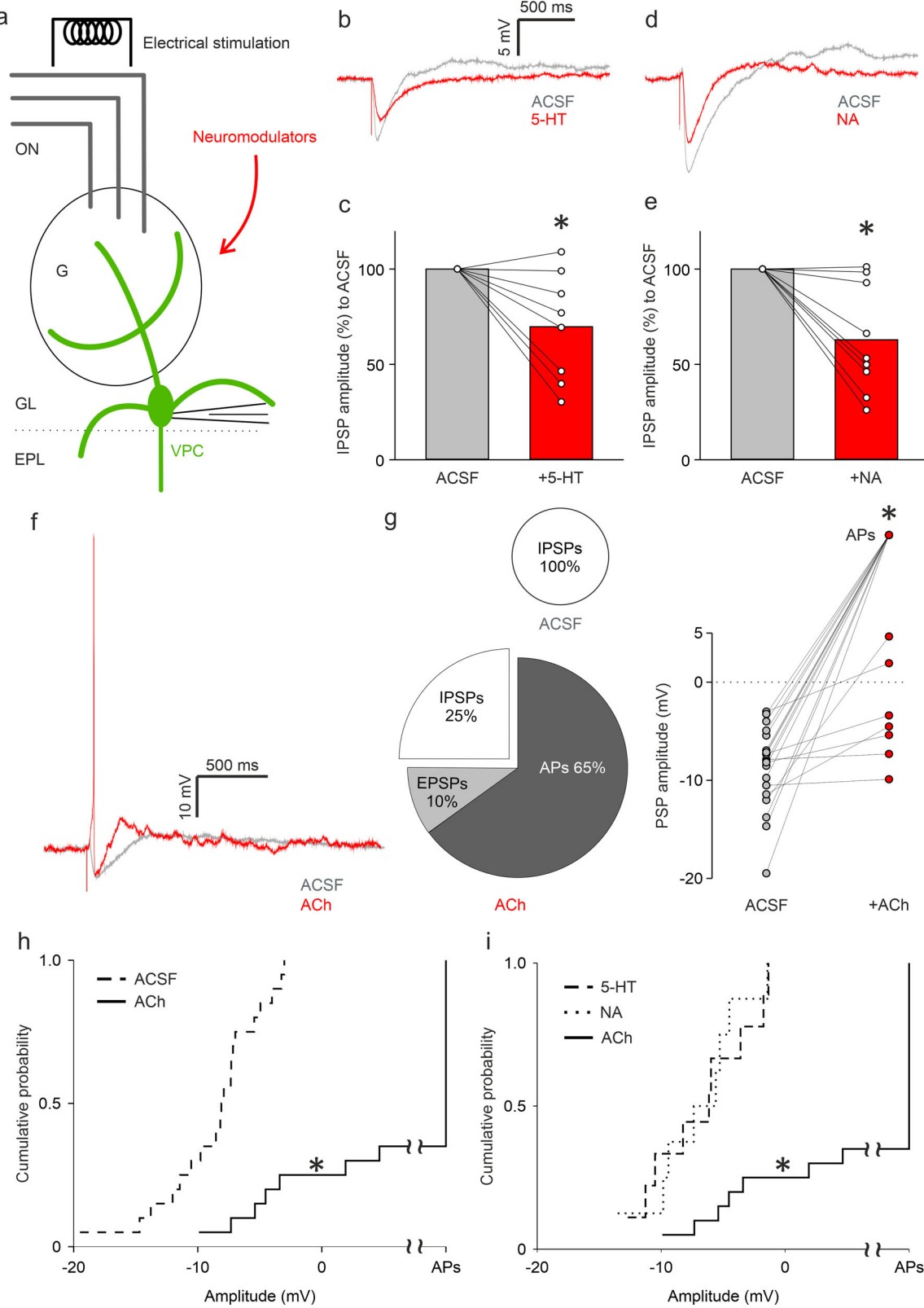

net response amplitude change towards positive/less negative values)$_i$. Accordingly, the distribution of response amplitudes in the presence of ACh was significantly different from control (Fig. 2h; $p < 0.001$)$_j$, as well as different from the 5-HT or NA condition (Fig. 2i; ACh vs. 5-HT $p = 0.003$, ACh vs. NA $p = 0.002$, 5-HT vs. NA $p = 1.0$)$_k$. These excitatory ACh effects were not due to an increased intrinsic excitability of VPCs since

ACh did not alter VPC AP spike frequencies during prolonged somatic current injections (Supplementary Fig. 2a + b; Treatment $p = 0.417$, Current intensity $p < 0.001$, Treatment*Current intensity $p = 0.528$, $n = 10$ from 9 rats)$_l$. Moreover, an increased latency of the first spike was observed for lower current injection levels, indicating rather reduced excitability during ACh treatment (Supplementary Fig. 2a + c; Treatment $p = 0.080$, Current

**Fig. 2 Cholinergic modulation triggers excitatory responses and action potentials during ON stimulation of VPCs. a** Schematic drawing of the experimental setup. Whole-cell patch-clamp recordings in 300 μm in vitro slices of responses from VPCs to electrical ON stimulation (50 μA, 100 μs, 30 s intervals). **b** Representative averaged traces of responses following ON stimulation in the ACSF condition (artificial cerebrospinal fluid, gray) and during bath application of 5-HT (serotonin 20 μM, red). **c** Cumulative analysis of normalized IPSP amplitudes of evoked IPSPs to the ACSF control condition during bath application of 5-HT ($n = 8$ cells). Data are presented as means including distribution of single data points. Lines between data points represent related measurements. Related-Samples Wilcoxon Signed Rank Test, *$p < 0.05$. **d** Representative averaged traces of responses following ON stimulation in the ACSF condition (artificial cerebrospinal fluid, gray) and during bath application of NA (noradrenaline 20 μM, red). **e** Cumulative analysis of normalized IPSP amplitudes of evoked IPSPs to the ACSF control condition during bath application of NA ($n = 9$ cells). Data are presented as means including distribution of single data points. Lines between data points represent related measurements. Related-Samples Wilcoxon Signed Rank Test, *$p < 0.05$. **f** Representative averaged traces of responses to ON stimulation in the ACSF condition (gray) and a single trace during bath application of ACh showing APs (acetylcholine, red). **g** Pie-chart represents the proportion of VPCs showing either APs, EPSPs or IPSPs in the ACSF and ACh condition. Dot-plots represent cumulative analysis of amplitudes of PSPs or APs (set as 100 mV) in the ACSF and ACh condition. Related-Samples Wilcoxon Signed Rank Test, *$p < 0.05$. $n = 20$ cells. (**h + i**) Cumulative probability of evoked PSP amplitudes in the ACSF, 5-HT, NA, or ACh condition ($n = 20/8/9/20$ cells). The amplitudes of APs were set as 100 mV. Kruskal–Wallis test for variation comparison, Bonferroni post-hoc, *$p < 0.05$ ACh vs. ACSF, 5-HT, NA. ON, olfactory nerve, G glomerulus, GL glomerular layer, EPL external plexiform layer, VPC vasopressin cell.

intensity $p < 0.001$, Treatment*Current intensity $p = 0.015$, ACh vs. ACSF (40 pA) $p = 0.029$)$_m$.

Rat AOB M/TCs send axons to the dorsal MOB[24], and it was shown that pheromonal AOB inputs are involved in social discrimination[21,25]. To investigate whether AOB inputs can elicit excitation in MOB VPCs, we electrically stimulated the AOB during whole-cell patch-clamp recordings from MOB VPCs. Acute in vitro slices for these experiments were prepared at a cutting angle that should preserve the respective projections[24] (Supplementary Fig. 2d). However, no VPC showed any responses (Supplementary Fig. 2e; $n = 21$), suggesting that modulation of MOB VPCs via AOB inputs is unlikely.

Taken together, we have identified ACh as a strong candidate neuromodulator that can unlock VPC APs and thus enable axonal VP release in the MOB.

**ACh modulates both ON-evoked inhibition and excitation but does not increase $Ca^{2+}$ influx into the apical dendrite.** For further dissection of the cellular underpinnings of these ACh effects, we quantified the amplitude changes of evoked IPSPs in VPCs that did not turn into APs or EPSPs in ACh (5 out of 20 cells, Fig. 2g). In these 5 cells, ACh decreased the amplitudes of evoked IPSPs to $67.8 \pm 11.3$ % of control (Fig. 3a + b; $p = 0.043$, $n = 5$ from 5 rats)$_n$. This observation indicates that ACh, like 5-HT and NA, also reduces inhibition of VPCs. We also examined the effect of ACh on the isolated excitatory components of ON-driven inputs in VPCs. After the detection of evoked IPSPs in ACSF, bicuculline, a GABA-A receptor antagonist, was applied to unmask evoked EPSPs[8], followed by ACh administration. Half of VPCs occasionally showed evoked APs under ACh application (Fig. 3d; 3 out of 6 cells from 5 rats). We averaged all subthreshold traces and found that ACh application increased the amplitudes of evoked EPSPs to $204.4 \pm 39.8$% from bicuculline alone (Fig. 3d; $p = 0.043$, $n = 5$ from 5 rats)$_o$. This increase in EPSP amplitudes indicates that ACh enhances ON-evoked excitation to activate VPCs, on top of reducing inhibition, possibly via acting on both excitatory and inhibitory glomerular neurons[26,27]. In addition, we analyzed the onset of ON-evoked APs in VPCs triggered by ACh modulation from Fig. 2 ($8.6 \pm 0.7$ ms; $n = 12$ from 12 rats, corresponding to Fig. 2g) and compared them to the onset of evoked EPSPs in bicuculline alone, which was significantly slower ($32.8 \pm 12.1$ ms; $n = 6$ from 5 rats, corresponding to Fig. 3c; $p = 0.002$)$_p$, further supporting the idea of additional excitation by ACh.

Since VPCs innervate glomeruli via apical dendritic tufts[8] which are known to receive sensory excitation in bulbar principal neurons, the MTCs[9], ACh-mediated excitation might act on processing within VPC tufts. To investigate subcellular processing

in VPCs, we simultaneously performed whole-cell patch-clamp recordings of PSPs from the soma and two-photon $Ca^{2+}$ imaging in the tufts and near the soma during ON stimulation (100 μM OGB-1, Fig. 3e + f). In the control condition, a single ON stimulation was able to elicit $Ca^{2+}$ influx in the tufts but not in the proximal apical dendrite near the soma of VPCs (Fig. 3f + g; tuft vs. soma $p = 0.002$, $n = 8$ from 8 rats)$_q$. However, upon ACh wash-in, although APs and EPSPs were recorded at the soma, ACh did not further increase $Ca^{2+}$ influx neither in the tufts nor the soma (treatment $p = 0.203$)$_q$. This observation implies that $Ca^{2+}$ influx at the soma is not responsible for the unlocking of somatic APs. Since 50 Hz-AP trains can elicit $Ca^{2+}$ signals also close to the soma in the proximal apical dendrites, the lack of such signals in response to single APs in the presence of ACh is not a technical artifact but probably due to a very low density of voltage-gated $Ca^{2+}$ channels in these compartments, as known from hypothalamic VPCs[28] (Fig. 3g; ON/soma vs. 50 Hz/soma $p = 0.014$; $n = 8$ from 8 rats)$_r$. Moreover, in the imaging experiments (with 100 μM OGB-1 in the internal solution) there was no difference in the effect of ACh on the increase of evoked PSPs compared to the previous electrophysiological experiments (Supplementary Fig. 3, $p = 0.456$; $n = 9$ vs. 11)$_s$ and therefore this effect is independent of the potential buffering of postsynaptic $Ca^{2+}$ by the $Ca^{2+}$ indicator. In conclusion, $Ca^{2+}$ influx into the tuft or proximal apical dendrite appears to be neither required for, nor modified by the ACh-mediated modulatory effects on ON-evoked PSPs.

**Muscarinic signaling is the major player in reducing inhibition of bulbar VPCs.** To narrow down mechanisms of cholinergic excitation of VPCs, we next focused on ACh receptor subtypes. The selective agonists (nicotine, 100 μM; muscarine, 1 μM) or antagonists (mecamylamine, a nicotinic receptor antagonist, 20 μM; atropine, a muscarinic receptor antagonist, 10 μM) were bath-applied to examine changes of ON-evoked PSPs in VPCs. Unlike ACh application, nicotine application was not able to turn evoked IPSPs into excitation in any of the tested VPCs (Fig. 4b; $n = 8$ from 5 rats). Moreover, the average amplitude of evoked IPSPs in nicotine was similar to control, even though variability across experiments was high (Fig. 4a + b; $p = 0.327$)$_t$. Similar results were observed with ACh application following blockade of muscarinic signaling with atropine. Neither application of atropine alone nor the atropine-ACh condition showed any evoked excitation and/or reduction of evoked-IPSP amplitudes (Fig. 4c + d; $p = 0.156$, Friedman test, $n = 7$ from 4 rats)$_u$. Thus, nicotinic signaling is most likely not sufficient to turn evoked inhibition to excitation in VPCs.

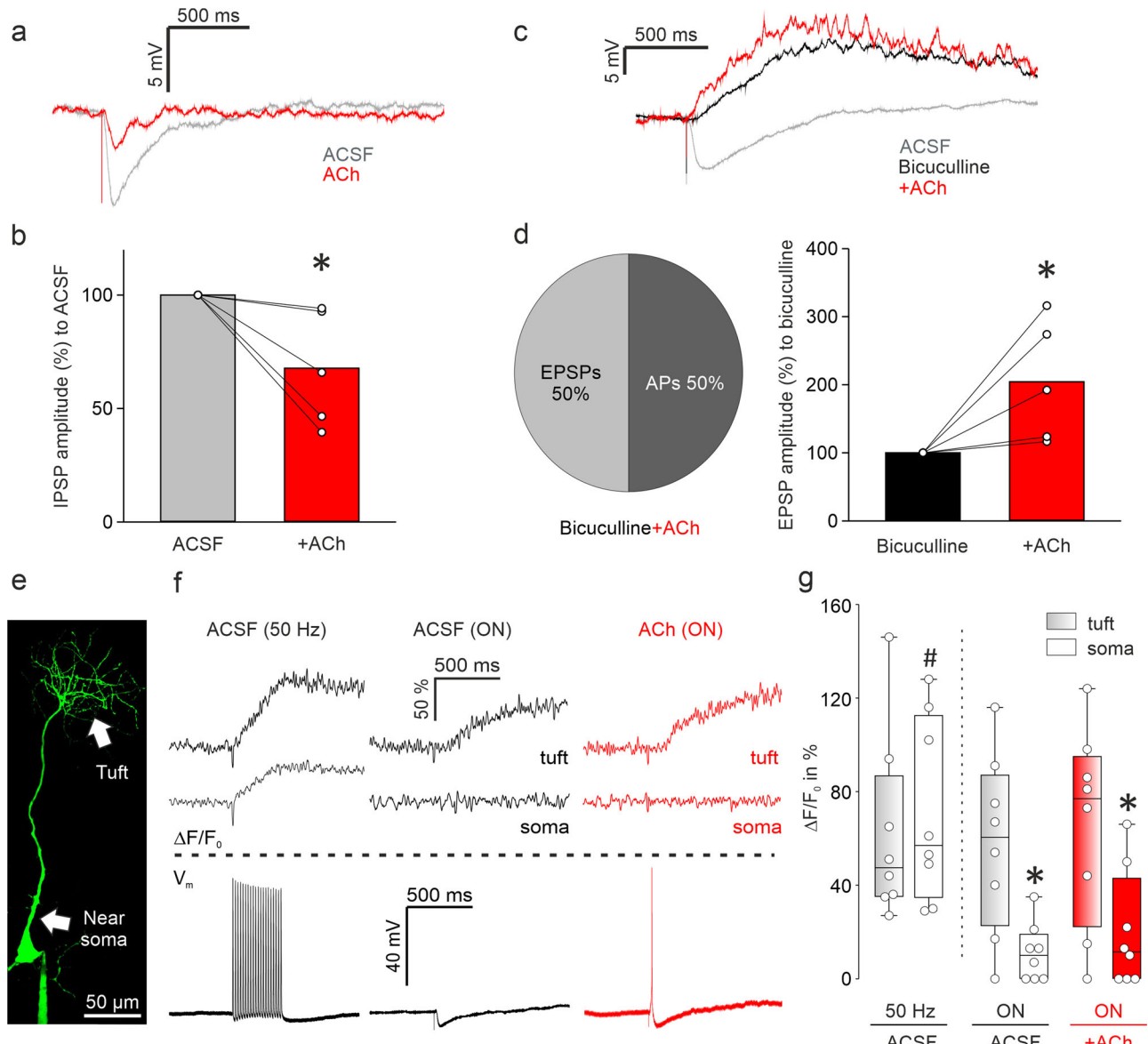

**Fig. 3 ACh modulates both ON-evoked inhibition and excitation via muscarinic receptors but does not increase intracellular Ca²⁺ influx.** Whole-cell patch-clamp recordings in 300 μm in vitro slices of responses from VPCs to electrical ON stimulation (50 μA, 100 μs, 30 s intervals). **a** Representative averaged traces of responses following ON stimulation in the ACSF condition (gray) and during bath application of ACh (acetylcholine, red) showing reduced IPSPs. **b** Cumulative analysis of normalized evoked IPSP amplitudes to the ACSF condition during bath application of ACh ($n = 5$ cells). **c** Representative averaged traces of responses following ON stimulation in the ACSF condition (gray) and during bath application of bicuculline (50 μM, black) and additional application of ACh (100 μM, red). **d** Pie-chart represents the proportion of cells showing either APs or EPSPs in the ACh condition ($n = 6$ cells). Bar-charts represent cumulative analysis of normalized evoked EPSP amplitudes to the bicuculline condition during additional application of ACh ($n = 5$ cells). Data are presented as means including distribution of single data points. Lines between data points represent related measurements. Related-Samples Wilcoxon Signed Rank Test, *$p < 0.05$ vs. ACSF/Bicuculline. **e** Representative picture of a VPC filled with OGB-1. Arrows indicate the tuft and the apical dendrite near the soma. Scale bar, 50 μm. **f** Representative averaged traces of Ca²⁺ influx in the tuft and the soma above the dotted line. Representative averaged traces of responses following ON stimulation below the dotted line (ACSF with 50 Hz somatic stimulation or ON stimulation, black; ACh with ON stimulation, red). **g** ΔF/F₀ in % from baseline following different stimulation and pharmacology. Data are presented as box-plots including first, median, and third quartiles with whiskers representing the range of data points and distribution of single data points. (2) × (2) mixed model ANOVA (location [within subject] × treatment [within-subject]) with treatment being either 50 Hz vs. ON or ACSF vs. ACh, *$p < 0.05$ vs. tuft within the condition. $n = 8$ cells. #$p < 0.05$ 50 Hz in the ACSF (soma) vs. ON in the ACSF (soma). $n = 8/8$ cells.

Conversely, muscarine administration enabled excitation in 25% of VPCs (Fig. 4f; APs and EPSPs in 1 cell each, $n = 8$ from 5 rats). Furthermore, muscarine application significantly reduced the amplitudes of ON-evoked IPSPs in the remaining 75% of cells (Fig. 4f; $p = 0.028$, $n = 6$ from 4 rats)ᵥ. Intriguingly, application of mecamylamine followed by ACh, which isolates muscarinic signaling, also resulted in ON-evoked excitation in a similar

amount of VPCs as during muscarine application (Fig. 4g+h; APs 18% and EPSPs 9% of cells, $n = 11$ from 9 rats). In addition, ACh following mecamylamine reduced evoked-IPSP amplitudes compared to control or mecamylamine alone (Fig. 4h; $p = 0.034$, Friedman test, mecamylamine-ACh vs. ACSF $p = 0.024$, mecamylamine-ACh vs. mecamylamine $p = 0.024$, 73 % of cells, $n = 8$ from 7 rats)ᵥ similar to muscarine application. Thus, these

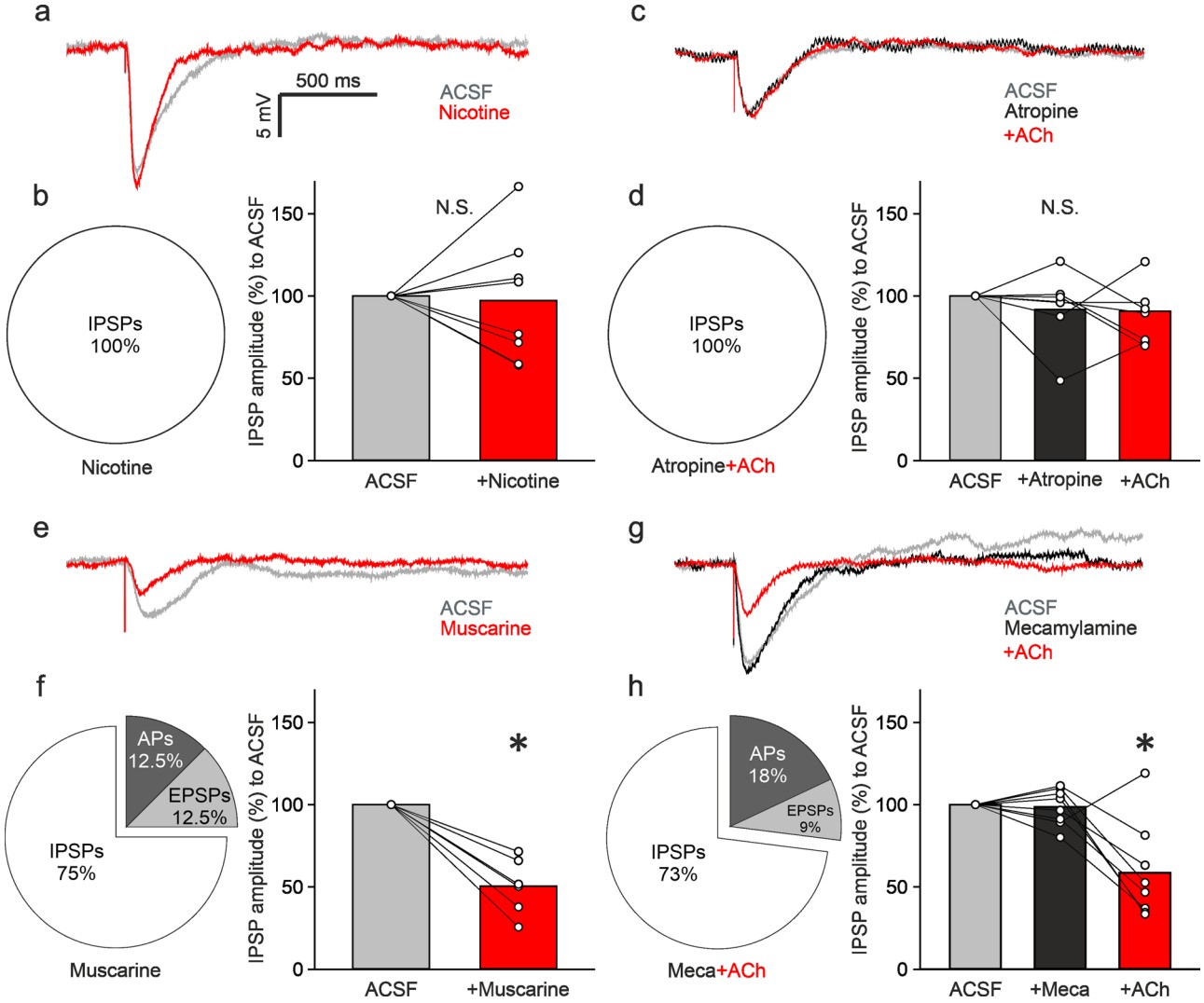

**Fig. 4 The muscarinic pathway is responsible for reduction of ON-evoked inhibition. a** Representative averaged traces of responses from VPCs following ON stimulation in the ACSF condition (gray) and during bath application of nicotine (100 μM, red). **b** Pie-chart represents the proportion of cells showing either APs, EPSPs, or IPSPs ($n = 8$ cells). Bar-charts represent cumulative analysis of normalized evoked IPSP amplitudes to the ASCF condition ($n = 8$ cells without APs and EPSPs). **c** Representative averaged traces of responses from VPCs following ON stimulation in the ACSF condition (gray) and during bath application of atropine (10 μM, black) and atropine+ACh (10 μM+100 μM, red). **d** Pie-chart represents the proportion of cells showing either APs, EPSPs or IPSPs in the atropine+ACh condition ($n = 11$ cells). Bar-charts represent cumulative analysis of normalized evoked IPSP amplitudes to the ASCF condition ($n = 8$ cells without APs and EPSPs). **e** Representative averaged traces of responses from VPCs following ON stimulation in the ACSF condition (gray) and during bath application of muscarine (1 μM, red). **f** Pie-chart represents the proportion of cells showing either APs, EPSPs or IPSPs in the ($n = 8$ cells). Bar-charts represent cumulative analysis of normalized evoked IPSP amplitudes to the ASCF condition ($n = 6$ cells without APs and EPSPs). **g** Representative averaged traces of responses from VPCs following ON stimulation in the ACSF condition (gray) and during bath application of mecamylamine (20 μM, black) and mecamylamine+ACh (20 μM+100 μM, red). **h** Pie-chart represents the proportion of cells showing either APs, EPSPs or IPSPs in the mecamylamine+ACh condition ($n = 7$ cells). Bar-charts represent cumulative analysis of normalized evoked IPSP amplitudes to the ASCF condition ($n = 7$ cells without APs and EPSPs). Data are presented as means including distribution of single data points. Lines between data points represent related measurements. Related-Samples Wilcoxon Signed Rank Test, $*p < 0.05$ vs. ACSF in agonist experiments. Friedman test, Dunn for single comparisons, $*p < 0.05$ vs. Meca (mecamylamine) or vs. ACSF in antagonist experiments. N.S., not significant.

results indicate that the muscarinic pathway is the major player in reducing inhibition of ON-stimulated VPCs, even though we cannot entirely exclude the involvement of nicotinic signaling.

**Cholinergic cells in the horizontal limb of the diagonal band of Broca are activated during social interaction.** Our results reveal that ACh can flip ON-evoked inhibition of bulbar VPCs into excitation in vitro. If this activating drive of ACh is indeed involved in triggering VPC excitation in behaving rats, we hypothesized that ACh should be released into the OB during

social interaction. Thus, ACh neurons that are known to project to the OB from the horizontal limb of the diagonal band of Broca (HDB)[29,30] should get activated by social interaction. Activity of the cholinergic system is correlated with changes in sensations or internal states such as arousal or attention[31–36]. Moreover, it was shown that stimulation of HDB ACh projections into the MOB can sharpen the odor responses of M/TCs[37,38]. Since we had obtained whole brains from a subset of the exposure experiments (water, rat urine, conspecific), we next investigated the activity of ACh neurons in the HDB. To visualize activated HDB neurons and specifically activated ACh neurons, we stained the HDB

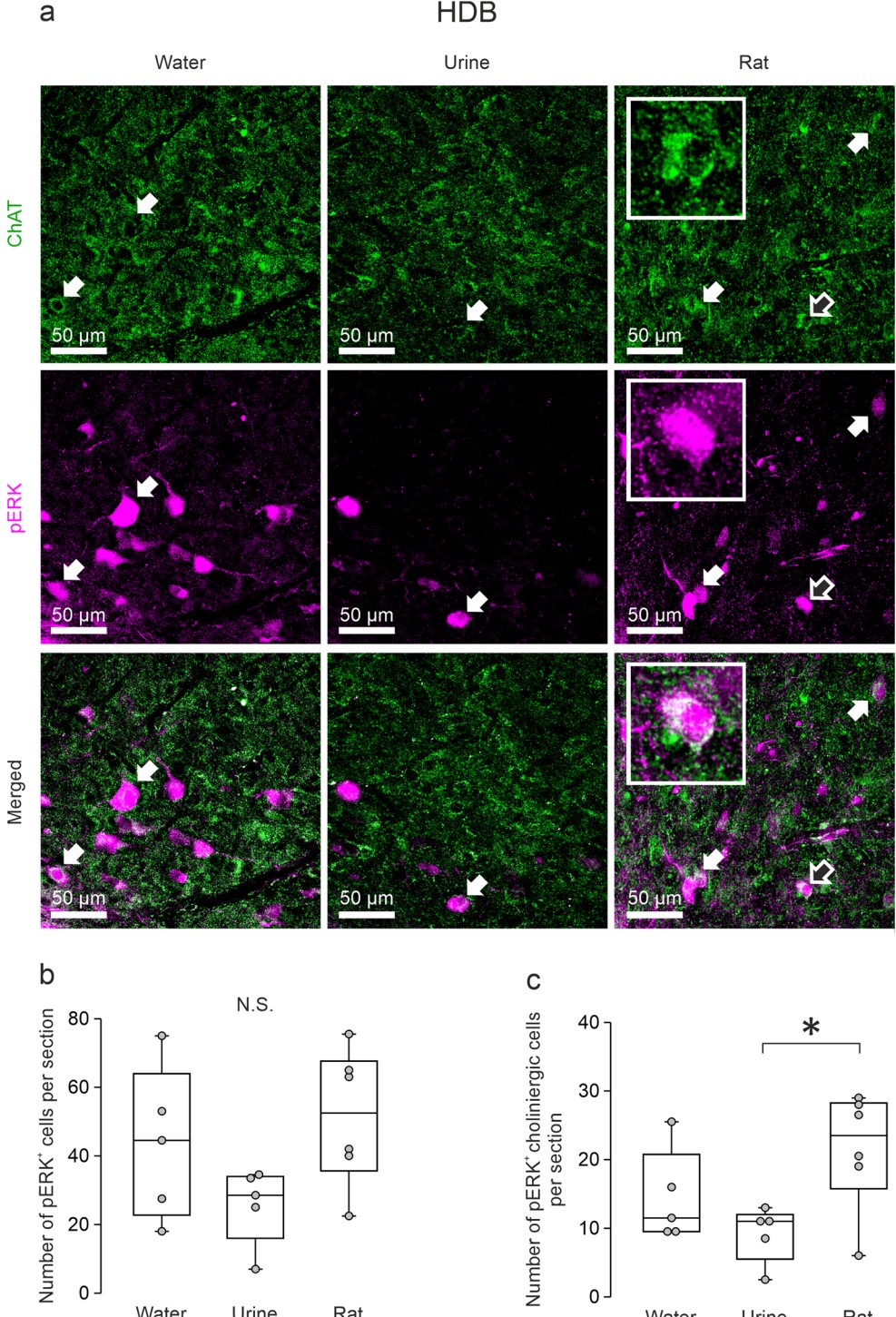

**Fig. 5 Cholinergic cells in the horizontal limb of the diagonal band of Broca are activated during social interaction. a** Representative average z-projections of the horizontal limb of the diagonal band of Broca (HDB) that were immune-stained for ChAT (choline-acetyltransferase, green, CF488) and pERK (magenta, CF 594) in the different experimental groups. Arrows indicate cells that are double-labeled for pERK and ChAT. Insets show enlarged pictures of the cell indicated by blank arrows. Scale bar, 50 μm valid for all images in the same panel. **b** + **c** Averaged number of pERK⁺ cells or pERK⁺ cholinergic cells per section in different experimental groups. Data are presented as box-plots including first, median, and third quartiles with whiskers representing the range of data points and distribution of single data points. One-way ANOVA, LSD for single comparisons, N.S., not significant, *$p < 0.05$ rat vs. urine, $n = 5$ rats (water), $n = 5$ rats (urine), $n = 6$ rats (rat).

against pERK and choline acetyltransferase (ChAT, Fig. 5a). The total number of pERK⁺ HDB cells was not significantly different between all groups (Fig. 5b; $p = 0.104$, ANOVA, water $n = 5$, urine $n = 5$, rat $n = 6$)ₓ. However, the number of pERK⁺ ACh

neurons in the rat exposure group was significantly higher than that in the urine exposure group, even though the pERK⁺ ACh neuron number in the rat exposure group was not higher than that in the water exposure group (Fig. 5c; $p = 0.034$, ANOVA, rat

vs. urine $p = 0.011$, rat vs. water $p = 0.113$, water vs. urine $p = 0.255$)$_y$. The results indicate that HDB ACh neurons are activated more easily by social interactions than by investigating urine and thus more likely to excite VPCs specifically during social interactions.

**Atropine-induced impairment of social discrimination is rescued by VP microinjection into the OB in male rats.** Our data so far demonstrated a disinhibitory effect of bulbar ACh on ON-evoked VPC activity mainly via muscarinic receptors as well as the activation of both systems, intrinsic bulbar VP and HDB ACh, during social interaction. These observations are in line with behavioral studies that demonstrate that pharmacological blockade of bulbar V1a receptor or systemic ACh activity diminishes social discrimination abilities in rats[7,39]. To examine if the cellular mechanisms we observed in the OB indeed play a role for social discrimination in behaving rats, we tested (1) if the blockade of muscarinic ACh signaling in the OB impairs social discrimination and (2) if a later additional VP injection, mimicking potential ACh-facilitated VP release, can rescue a possibly impaired behavior. Thus, we performed microinjections of atropine (1 μg/ 1 μL per hemisphere) or vehicle (ACSF, 1 μL per hemisphere) followed by either VP (1 ng/ 1 μL per hemisphere) or vehicle before the sampling phase of a social discrimination test (Fig. 6a). In this test, the ability to discriminate two individuals is measured in terms of how long rats investigate a novel versus a known stimulus rat[4].

Control rats that received only vehicle investigated the novel stimulus rat significantly more than the known rat, indicating intact social discrimination (Fig. 6d; $p = 0.033$, $n = 13$)$_z$. In contrast, atropine-injected rats showed a similar duration of investigation towards both, novel and known stimulus rats (Fig. 6e; $p = 0.240$, $n = 14$)$_{a1}$. Thus, the blockade of muscarinic signaling in the OB indeed impaired social discrimination. Atropine-VP injected rats, however, significantly preferred investigating the novel stimulus rat over the known one (Fig. 6f; $p = 0.031$, $n = 14$)$_{b1}$. Hence, impairment of social discrimination by atropine is most likely caused by a disruption of the activation of the bulbar VP system, and in turn a reduced bulbar VP release during social interaction. Therefore, their ability to discriminate a known from a novel stimulus was rescued by additional VP injection.

To confirm that neither atropine nor VP microinjection into the OB induced unspecific behavioral effects, we demonstrated that these manipulations did not interfere with play behavior (Fig. 6g; $p = 0.633$, $n = 41$)$_{c1}$ and habituation to the juvenile rat (Fig. 6h; $p < 0.001$, $n = 41$)$_{d1}$ during the sampling phase. Furthermore, also non-social investigatory behavior (Supplementary Fig. 4a; $p = 0.843$, $n = 41$)$_{e1}$ and habituation towards amyl acetate or (-)-carvone presented in a teaball (Supplementary Fig. 4b; $p < 0.001$, $n = 41$)$_{f1}$ was not changed by the pharmacological treatments.

This final experiment thereby supports our hypothesis based on our findings in vitro that the activity of the bulbar VP system (that is essential for social discrimination) is triggered by centrifugal ACh inputs during social interaction.

## Discussion

Social discrimination in rats depends on the presence of endogenous VP in the OB[7], which implies that social interaction can trigger VP release, via the activation of bulbar VPCs. A substantial background activity of VPCs (pERK) was found throughout all stimulation groups in vivo, even though VPCs are inhibited by olfactory nerve input in vitro. This background activity might be related to the circumstance that it is not realistic to deprive our experimental rats from all environmental sensory

inputs, e.g., rat odors from other cages[40]. Nevertheless, we observed increased levels of bulbar VPC activation during social interaction (Fig. 1c + d). Thus, our results suggest that during social interaction VPC activity and hence the probability of bulbar VP release, are substantially increased (Fig. 7). This finding is apparently at variance with others from Wacker et al.[41], who reported that social interaction cannot trigger the expression of the immediate early genes c-Fos or Egr-1 in bulbar VPCs. However, as discussed by them, a missing expression signal of those genes might be explained by their general absence of expression in bulbar VPCs.

Since we know that VPCs are primarily inhibited by ON inputs in vitro[8], we hypothesized that additional inputs from outside of the OB are responsible for the activation of VPCs during social interaction.

Pheromones from conspecifics are detected by rodents during social interactions and were shown to directly influence social behaviors, such as sexual behavior or aggression between conspecifics[42,43]. Thus, pheromones would be an obvious way to provide social specificity via social interactions. Indeed, M/TCs in the AOB, the first pheromonal relay station in the brain, are known to innervate the dorsal MOB and therewith could also excite bulbar VPCs[24]. In our hands, electric stimulation of the AOB failed to elicit any excitatory responses in MOB VPCs in in vitro sagittal slice experiments (Supplementary Fig. 2d + e). Moreover, there was no significantly enhanced MOB VPC activation by urine, containing pheromones, compared to control. However, we cannot entirely rule out this possibility because of the reduced connectivity in vitro and the larger/different set of pheromonal cues accessible during investigation of a conspecific versus just urine. Moreover, although social interaction increases c-Fos expression in the AOB in general[25], we did not find any differences in the number of pERK$^+$ AOB M/TCs between the rat, urine, and water exposure group (Supplementary Fig. 1c+f). Also, we rarely observed pERK$^+$ VPCs in the AOB itself, neither following urine exposure nor social interaction (only 1 or even less cells on average per slice). In summary, pheromonal AOB signaling is rather unlikely to underlie VP-dependent individual social discrimination in the MOB.

Also, while removal of the vomeronasal organ initially prevents social recognition of same sex individuals, it is reinstated after 14 days, indicating that AOB functioning is not required for social recognition per se[21]. Nevertheless, the AOB is crucial for individual discrimination in the context of reproduction. For example, the pheromonal memory of the stud male in female mice results in a pregnancy block induced by odors of male strangers. This consequence of social odor recognition, known as the Bruce effect, depends on pheromonal processing in the AOB[44]. Thus, the AOB may be essential for individual discrimination in certain situations, such as in a reproductive context, but not necessarily in social discrimination in general. Still, it is possible that modulatory top-down inputs to the MOB that are activated by AOB processing[45] are involved in the activation of MOB VPCs.

Cholinergic neurons in the HDB are the only known source that provides centrifugal neuromodulatory inputs to the GL of the OB, where VPCs are predominantly located[8,29,30]. Following social interaction, the number of pERK$^+$ cholinergic HDB neurons was significantly higher compared to rat urine exposure (Fig. 5c) indicating that cholinergic activation might be involved in bulbar VPC activation in the rat exposure group (Fig. 1c). Accordingly, we observed that ON-evoked IPSPs are reversed into EPSPs/APs in 75% of examined VPCs during ACh administration in vitro (Fig. 2g). Moreover, we showed that ACh induces both, reduced inhibition, and stronger ON-driven excitation (Fig. 3a–d, Fig. 7). Reduced GABAergic inhibition by ACh

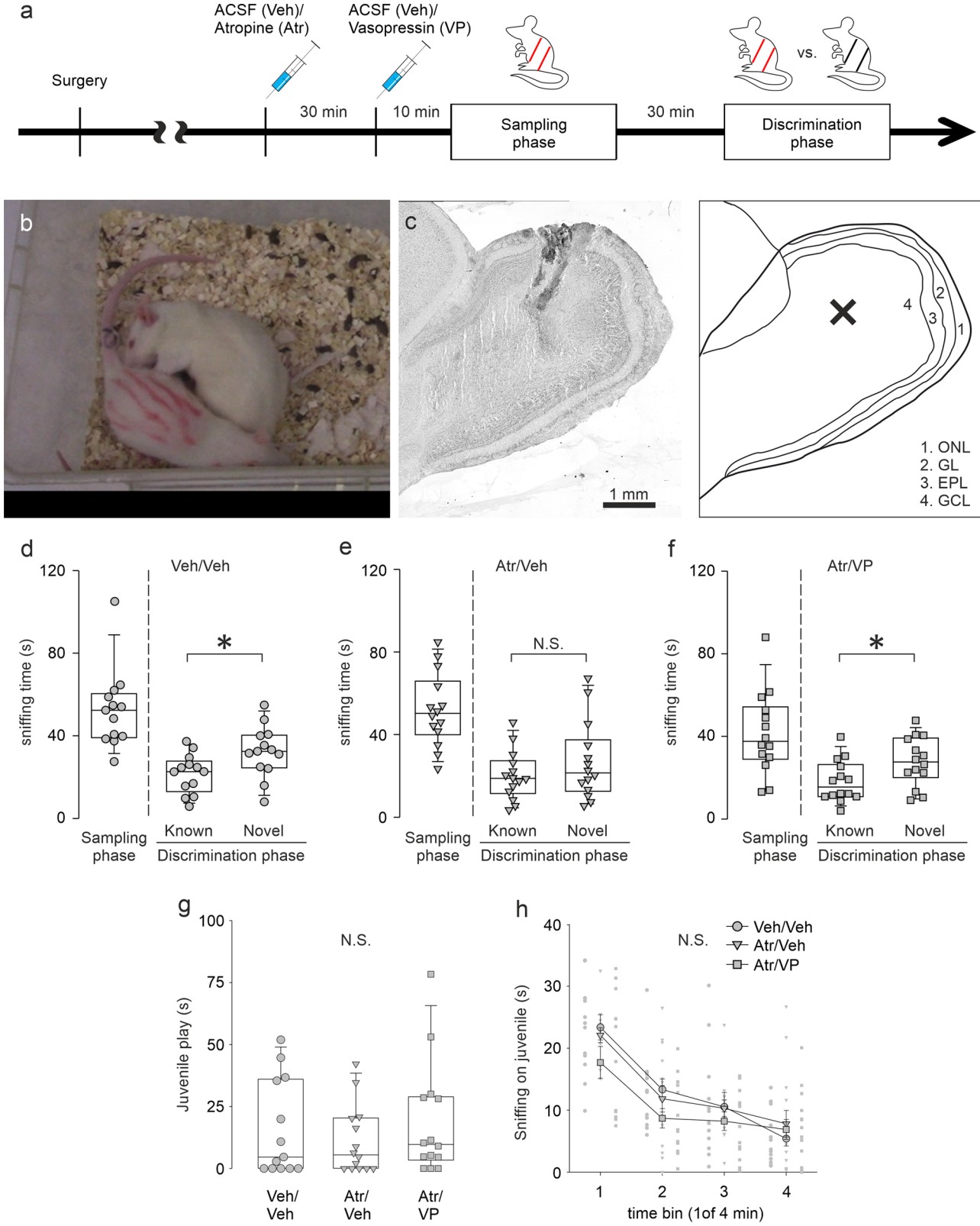

was already observed in hypothalamic VPCs, resulting in increased VP release in vitro[46,47].

Non ACh-specific optogenetic silencing of all vesicular release from HDB projections in the OB reduces glomerular responses to odor stimulation in vivo[30]. In addition, optogenetic stimulation of specifically HDB cholinergic projections in the OB in vivo increases odor responses in projection neurons[37,38]. Both results indicate overall excitatory effects of HDB cholinergic modulation in the OB during odor stimulation. Cells in the GL were reported to have a particularly high sensitivity for ACh, and direct application of ACh resulted in both, increase and decrease of spontaneous spiking rates in individual cells there[48]. These bidirectional effects may be explained by the presence of two different ACh receptor subtypes, nicotinic and muscarinic

**Fig. 6 VP microinjection into the OB rescues atropine-induced impairment of social discrimination in male rats. a** Schematic time course of the experimental design. **b** Rat (5 weeks) without markings representing anogenital sniffing toward a stimulus rat (3 weeks) with red markings. **c** Representative picture of the OB with dye injection via drug injection system, counterstained with Cresyl violet and a schematic picture of the olfactory bulb. The cross in the schematic picture indicates the approximate injection position in the left picture. ONL olfactory nerve layer, GL glomerular layer, EPL external plexiform layer, GCL granule cell layer. **d–f** Amount of time (s) that rats investigate stimulus rats in the sampling phase and in the discrimination phase (known or novel rat) following microinjection (1 µL per bulb) with only vehicle, atropine (1 µg/ 1 µL) and vehicle or atropine and VP (1 ng/ 1 µL). Data are presented as box-plots including first, median, and third quartiles with whiskers representing the range of data points and distribution of single data points. Paired Samples $t$ test between known and novel, *$p < 0.05$, $n = 13$ rats (Veh/Veh), $n = 14$ rats (Atr/Veh), $n = 14$ rats (Atr/VP). **g** Amount of time (s) that rats are engaged in playing with the stimulus rat during the sampling phase. Data are presented as box-plots including first, median, and third quartiles with whiskers representing the range of data points and distribution of single data points. Kruskal–Wallis Test, $n = 13$ rats (Veh/Veh), $n = 14$ rats (Atr/Veh, 1 µg), $n = 14$ rats (Atr/VP, 1 µg/1 ng). **h** Amount of time (s) within time bins of 1 min that rats investigate the stimulus rat during the sampling phase following different drug applications. Data are presented as means ± SEM including distribution of single data points. (4) × (3) mixed model ANOVA (time bin [within subject] × treatment [between-subject]), $n = 13$ rats (Veh/Veh), $n = 14$ rats (Atr/Veh, 1 µg), $n = 14$ rats (Atr/VP, 1 µg/1 ng). N.S., not significant.

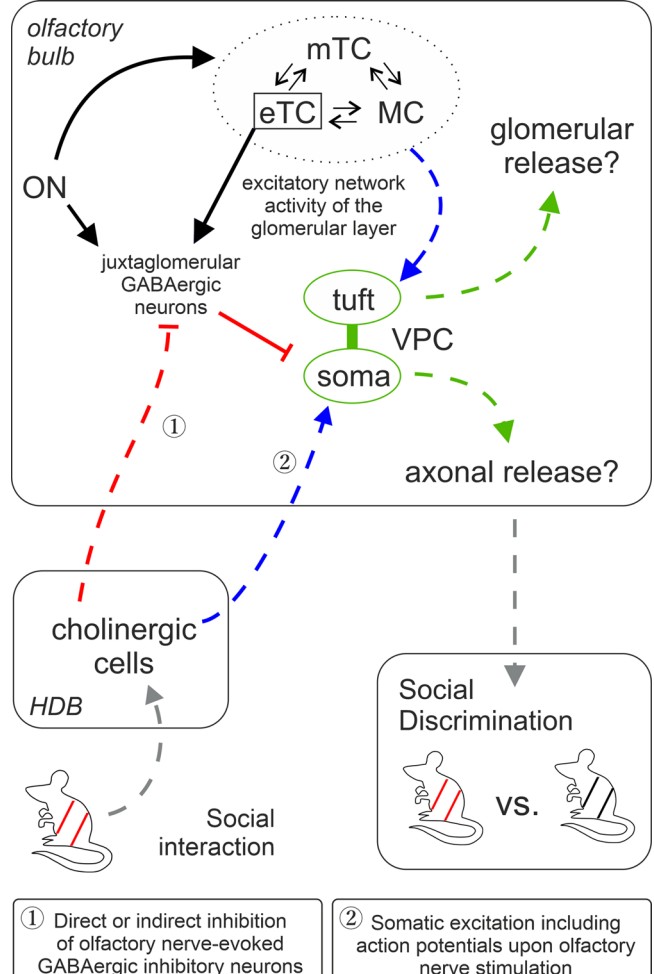

**Fig. 7 Graphic summary of OB pathways involved in social discrimination.** Blue, green, and gray arrows represent excitatory and red lines inhibitory pathways indicated by this study and our previous publication[8]. The cells within the dashed circle indicate the excitatory network within the same home glomerulus. Dashed pathways were confirmed but not fully dissected on the synaptic level. Green pathways are related to vasopressin signaling, gray pathways to sensory input or behavioral output and black pathways are based on findings from literature. Question marks indicate potential dendritic and axonal release of VP in the OB. Axonal projections innervate GL, EPL, MCL, and sGCL. EPL external plexiform layer, eTC external tufted cell, GL glomerular layer, HDB horizontal limb of the diagonal band of broca, MC mitral cell, MCL mitral cell layer, mTC middle tufted cell, ON olfactory nerve, sGCL superficial granule cell layer, VPC vasopressin cell.

receptors, and their distinct effects on the various cell types of the OB[49].

In our experiments, we demonstrated that muscarinic stimulation with either muscarine or mecamylamine-ACh application reversed evoked inhibition into excitation in some VPCs (Fig. 4e–h). At the level of the GL, it was shown that the muscarinic pathway enhances inhibitory inputs to GABAergic interneurons, i.e., periglomerular cells and short axon cells, hence possibly causing disinhibition of VPCs[26]. According to our data, mainly muscarinic but not nicotinic stimulation reduced ON-evoked IPSP amplitudes in VPCs (Fig. 4a–h). In line with our observation, in vivo $Ca^{2+}$ imaging revealed that muscarinic stimulation increases glomerular sensitivity to odor presentation[50].

All our pharmacological manipulations in vitro were performed via bath application. Since nicotinic receptors are ionotropic and known to get desensitized[51], locally and temporally more precise administration, such as puff application or optical uncaging, may be informative to investigate nicotinic mechanisms on a single cell level. Indeed, although we observed no statistical differences in nicotine effects on evoked-IPSP amplitudes, increasing and decreasing effects of nicotine application could be found across individual VPCs (Fig. 4a+b). In this context, D'Souza and Vijayaraghavan[52] described that nicotinic receptor signaling is involved in gain control mechanisms of excitatory projection neurons in the OB, which might explain the variable effects that we observed upon nicotine and atropine-ACh application. Moreover, muscarine application did not entirely replicate ACh effects (Fig. 2g, Fig. 4f+h). Thus, we cannot fully exclude nicotinic action in modulation of VPCs. Furthermore, the origin of the ON-driven excitation that is responsible for triggering APs during ACh application still needs to be identified. Possible candidates would be neurons providing somatic and lateral dendritic inputs to VPCs since ON-evoked $Ca^{2+}$ signals in the tuft were not increased in ACh (Fig. 3g).

Although here we focused on the cholinergic system because of the strength of cholinergic effects, our in vitro electrophysiology results indicate that increased VPC excitability could be triggered by synergistic cholinergic, serotonergic, and noradrenergic modulation; the possibility of such cooperative action should be accounted for in future studies.

If centrifugal cholinergic modulation from the HDB is indeed responsible for increasing VPC activity, the question remains how ACh is released in the OB in a context-dependent manner such as social interaction. Social perception is thought to be a multisensory process as diverse inputs from the whole stimulus animal are sampled and integrated[4,10,53]. In support of this idea, Matsuo et al.[3] showed that genetic ablation of dorsal MOB glomeruli impairs the expected emission of ultrasonic vocalization by male mice towards female urine, whereas these mice still emit ultrasonic vocalizations

in the presence of a whole female mouse. This indicates that other sensory stimuli than those provided by olfaction can trigger this social behavior. Moreover, it was shown that rats, which were exposed to just volatile odors of stimulus rats during the social discrimination paradigm cannot discriminate the different sources of these odors[25]. According to our hypothesis, this impairment might be due to the absence of non-olfactory cues and lack of arousal.

Intriguingly, the HDB or the ACh system in general is activated by various sensations, such as olfactory, visual, and tactile stimuli[31,32,36]. Stimulation of the HDB cholinergic projections or ACh application in turn can enhance as well as reduce neural activity in all primary sensory cortices[54–56]. Also, especially if the sensation is associated with learning, ACh tends to be released in the respective sensory cortex, for example during conditioning of auditory or tactile cues with food as an unconditioned stimulus[36,57]. Accordingly, our data demonstrates substantial HDB ACh activity during social interaction. Surprisingly, we were not able to find a significant difference between water stimulation and social interaction in ACh neurons of the HDB. It is not possible to determine from our data what may be the reason for this high activation in the water condition. However, water stimulation (with no or few olfactory cues) was not enough to increase pERK activity in MOB VPCs to the same level as social interaction, suggesting that both, ACh neuron activation in the HDB and olfactory inputs are needed for VPC activation. In support, our in vitro electrophysiology results demonstrate that ACh modulation promotes especially excitation in VPCs following ON stimulation mimicking odor stimulation. Thus, it is tempting to speculate that the HDB cholinergic system could play a role as a multisensory integration center during complex contexts like social interaction that provides modulatory feedback to sensory processing, including but not limited to olfaction. Intriguingly, similar multi-sensory integration is also discussed to take place in the insular cortex which is responsible for coordinated behavioral output during social decision-making[58]. From a translational point of view, it is interesting to note that also human research on social behavior deficits suggests a social-context network that integrates social cues (frontal lobe), consolidates social-context associations (temporal lobe), and converges environmental and internal signals (insular lobe)[59].

Moreover, ACh activity correlates with certain internal states in animals, e.g., arousal or attention[33–35]. This internal state-dependent ACh release is triggered by the sudden presence or novelty of stimuli[16,60]. Rats discriminate individuals according to their social novelty[4]. Thus, we suggest that both multi-sensation and increased arousal and attention while assessing the novelty of rats in the context of social interaction can lead to increased HDB ACh activity and thereby cholinergic neurotransmission in the OB (Fig. 7).

Our pERK and electrophysiological experiments imply a central role for ACh in VP-dependent social discrimination and accordingly social memory. The involvement of centrifugal ACh as a facilitator of olfactory learning or habituation has been investigated for decades[61]. The blockade of the muscarinic pathway in the OB impairs both aversive and appetitive learning[62,63]. Intriguingly, it was shown that the blockade of muscarinic neurotransmission in the OB by scopolamine does not impair single-molecule odor recognition[64]. However, non-ACh specific optogenetic silencing of overall release from HDB projections in the OB impairs social habituation to female conspecific mice[30]. In addition, here we demonstrated that the injection of the muscarinic antagonist atropine in the OB efficiently blocks the recognition of a known rat in the social discrimination task. These findings indicate that social discrimination/habituation paradigms and simple odor discrimination/habituation paradigms differ from each other.

Regarding sensory complexity, social discrimination recruits multiple senses, whereas odor discrimination is limited to mostly mono-molecular odorants. As mentioned above, multisensory stimulation is a known trigger for the activation of HDB ACh neurons. Atropine-induced impairment of social discrimination was rescued by additional VP injection in the OB (Fig. 6f). Therefore, we suggest that ACh facilitates social discrimination via enhancing bulbar VP release. Since bulbar VP signaling is essential for multisensory social discrimination but not for simple odor habituation[7], we suggest that the probability of bulbar VP release is higher during social discrimination. This hypothesis is supported by a study with another social behavior-related neuropeptide, oxytocin, in which Martinon et al.[65] demonstrated that oxytocin release in the bed nucleus of stria terminalis is not increased with a foot shock alone, but when the foot shock is paired with another sensory cue (condition stimulus). Intriguingly, multisensory integration was recently also implied in a context of social discrimination. Accordingly, rats and mice cannot recognize their cage mates if they are immobile due to anesthesia, and mice that are both whiskerless and deaf are not able to discriminate awake novel and known conspecifics anymore[66].

We showed that social interaction as well as ACh activate bulbar VPCs, indicating that cholinergic activation contributes to increased activation of the bulbar VP system during social interaction. However, the pERK results also indicate that the entire neuronal population of the bulb shows increased activation, not just VPC neurons, upon social exposure. Thus, alternatively to a multisensory driven specific activation of bulbar VPCs, it is possible that a general arousal mechanism leads to an increased excitation of the entire bulbar network, including VPCs. Accordingly, this might also explain to a certain degree why urine, in contrast to a rat stimulus, did not activate significantly more VPCs than water, since urine might cause less general arousal than a hitherto unknown conspecific.

The absence of sensitivity to both, the muscarinic antagonist or a V1a receptor knock-down by siRNA in simple odor discrimination and habituation[7,64] implies that ACh-induced VP signaling is only necessary to differentiate difficult odor mixtures. Interestingly, ACh was reported to play a critical role in modulating olfactory perceptual learning tasks[61]. Perceptual learning is a prerequisite for enhanced perceptual acuity during discrimination of previously experienced stimuli. Like social discrimination, perceptual learning is specific for trained stimuli (e.g., a conditioned odor, a previously sampled rat) and requires a certain attention/arousal of the learning animal. This enhancement of olfactory acuity is mediated by muscarinic signaling[61]. Thus, ACh-dependent VP neurotransmission might be involved not only in social discrimination, but also olfactory perceptual learning tasks in general, e.g., discrimination of closely related odor mixtures. Indeed, different body odors should be difficult to discriminate due to their complexity and similarity. Singer et al.[67] showed that volatile MHC components in mouse urine contain similar compositions of compounds across strains suggesting that volatile MHCs are not distinguishable by the presence or absence of some unique compounds. However, the amplitudes of some prominent compound peaks in the gas chromatogram are different between strains, indicating that mice can recognize the differences of relative concentrations of certain compounds in body odor mixtures. Rats perceive not only volatile urinary odors but also volatile odors from other sources and even differentiate individuals from the same strain during the social discrimination task. Thus, it is likely that the olfactory cues that they can discriminate are even more complex. We therefore suggest that additional bulbar VP neuromodulation is needed to enhance the

signal-to-noise ratio of complex odor inputs to allow better identification of individual body odors. Reduced activity of individual projection neurons via increased activity of interneurons is thought to act as a filtering mechanism[68,69].

Although social interaction increased pERK activation in MCs (Fig. 1e), inhibition of pERK+ MCs, e.g., via a reduction of their firing frequency could still take place. E.g., pERK signaling could be not sensitive or fast enough to register this inhibition or pERK signaling is just gradually decreased but still higher than during baseline activity. Indeed, it was already shown that VP administration dampens odor-evoked firing of MCs in anesthetized rats[7] and reduces ON-evoked EPSPs in external tufted cells in vitro[8]. Thus, we suggest that this filtering effect facilitated by bulbar VP can further reduce weak or quantitatively similar inputs, so that only quantitatively prominent inputs are transmitted to higher brain regions leading to more prominent differences in the neuronal representation of different social odor signatures. This reduction of noise could help the rats to discriminate similar mixtures of body odors during the social discrimination task, but also during social interaction in nature.

## Methods

**Animals**. All experiments were conducted according to national and institutional guidelines for the care and use of laboratory animals, the rules laid down by the EC Council Directive (86/89/ECC) and German animal welfare. The study protocol was approved by the Government of Unterfranken (RUF-55.2.2-2532-2-539 and RUF-55.2.2-2532-2-1291). Wistar rats of either sex were purchased from Charles River Laboratories (Sulzfeld, Germany) or bred onsite in the animal facilities at the University of Regensburg. Heterozygous VP-eGFP Wistar rats[20] of either sex were bred at the University of Regensburg. The light in the rooms was set to an automatic 12 h-cycle (lights on 07:00-19:00).

### In vivo social stimuli exposure experiment

*Stimuli exposure*. Three cohorts of male VP-eGFP rats (5-6 weeks) were single-housed at least 3 h before the experiment for habituation to the new environment. Stimuli used were water, juvenile rat urine (collected and mixed from at least 3 juvenile male, stimulus rats) and juvenile rats (3-4 weeks, male non-cage mates). Water and urine (both 50 µl) were applied on filter papers (2 cm × 3 cm). The different stimuli were gently introduced in the cage and the rats could explore freely. A stimulus exposure lasts 4 min in which the behavior of the rats was video-taped for confirmation of proper stimulus sampling. Rats that did not directly sniff at the filter paper containing water or urine as well as rats that did not perform proper social investigation (sniffing anogenital and head region, for details see Video analysis: Social investigation) were excluded from the experiment. Immediately after the exposure, rats were deeply anesthetized for transcardiac perfusion and fixation of the brain.

*Transcardiac perfusion*. Rats were deeply anesthetized with an i.p. injection of ketamine-xylazine (100 mg/kg and 10 mg/kg, respectively). The abdomen and the diaphragm were incised to expose the thoracic cavity. After opening the mediastinum, a needle was inserted into the left ventricle. Following incision of the right atrium, 0.1 M PBS pH 7.4 was perfused for 4 min with a speed of 9.5 mL/min by a pump (GZ-07528-30, Cole-Parmer, Wertheim, Germany) followed by 4% PFA-PBS for 4 min. Rats were decapitated and the whole brains were extracted. Brains were post-fixed in 4% PFA-PBS overnight at 4 °C and stored in 30% sucrose-0.1 M PB at 4 °C for at least 2 days then kept at 4 °C until slicing.

*Immunohistochemistry*. Brains were sliced with a cryostat (CM3050 S, LEICA, Wetzlar, Germany) at approximately −20 °C, then stored in cryoprotectant solution (0.1 M phosphate buffer, 500 mL; Ethylene Glycol, 300 mL; Sucrose, 300 g; Polyvinyl-pyrrolidone, 10 g) at −20 °C until staining. The OB was cut horizontally with a thickness of 30 µm. On average, six slices from one bulb of each experimental rat were used for staining. The HDB (Figure 24-28 (1.08 − 0.60 mm from the Bregma) in the rat brain atlas[70]) was cut coronally with a thickness of 30 µm. Three slices of each experimental rat were used for staining.

All immunohistochemistry procedures were performed in 12 well plates (Corning Incorporated, Corning, NY, USA) with the free-floating method. Slices were washed three times with 0.3 % Triton-X100 in PBS (PBST) for 10 minutes. Then slices were incubated in methanol for 10 minutes at −20 °C. After washing with PBST for 10 minutes, incubation with 0.1 M Glycine in PBS was performed for 20 minutes at room temperature, followed by washing with PBST for 10 minutes. Slices were incubated with the blocking solution for one hour at room temperature. The blocking solution contained 0.2 % of cold water fish skin gelatin (AURION, Wegeningen, Netherlands) and 0.01 % of NaN₃ in normal donkey serum (NB-23-00183-1, NeoBiotech, Nanterre, France). Incubation with primary

antibodies (see Antibodies) diluted in the blocking solution was carried out for 72 hours at 4 °C. After washing three times with PBST for 10 minutes, secondary antibodies diluted in the blocking solution were added and incubated for two hours at room temperature, followed by washing three times with PBST for 10 minutes. From the incubation in secondary antibodies on, every procedure was performed in the dark to avoid bleaching of the fluorescence. For choline acetyltransferase (ChAT) staining, the protocol was modified. Additional blocking by incubation with avidin and biotin (both 0.001% in PB, Sigma-Aldrich, Darmstadt, Germany) for 30 min and a 10 min wash with PBST in between were carried out before incubation with the blocking solution. Moreover, after the last washing with PBST, incubation in streptavidin conjugated with CF488 (1:400 in PBST, #29034, Biotium, Fremont, CA, USA) for two hours at room temperature was carried out. After staining itself, slices were washed three times with PBST for 10 min. Slices were mounted on objective slides (Thermo Fisher Scientific, Waltham, MA, USA) using DAPI fluoro mount-G (Southern Biotech, Birmingham, AL, USA).

*Antibodies*. Primary antibodies were goat anti-GFP (1:1000, #600-101-215 S, Rockland, Limerick, PA, USA), Rabbit anti-P-p44/42 MAPK (1:1000, #9101 S, Cell Signaling Technology, Frankfurt am Main, Germany), Sheep Anti-Choline Acetyltransferase (1:125, ab18736, Abcam, Berlin, Germany). Secondary antibodies were Donkey anti-Goat IgG conjugated with CF488 (for GFP, 1:1000, #20016-1, Biotium), Donkey anti-Rabbit IgG conjugated with CF594 (for pERK, 1:1000 for the OB and 1:500 for the HDB, #20152, Biotium), biotinylated Donkey anti-Sheep IgG (for ChAT, 1:250, #20191-1, Biotium). All antibodies were diluted in the blocking solution.

All purchased antibodies were validated for target and species specificity as indicated in the data sheets of the respective companies' websites.

**Fluorescent microscopy**. Fluorescent images of the stained slices were obtained using a DM6 B microscope (LEICA) and the software, LAS X (LEICA). DAPI, CF488 and CF594 were stimulated with an exposure time of 300 ms, 300 ms and 500 ms, respectively. Tile-stitching and z-stack of pictures were performed by LAS X. After taking pictures, those were processed by 3D deconvolution or the Lightning & Thunder process (LAS X) to improve the contrast. Pictures were converted to .tif files using Fiji (ImageJ, downloaded from https://imagej.net/Fiji/Downloads) and contrast was adjusted to the same levels in every picture. Z-stack pictures from ~5–6 different z-positions per bulb or rats were used for analysis.

**Cell counting**. Immunoreactive cells were counted manually using the multi-point tool or cell counter plug-in in Fiji. 4-6 OB slices and 2 HDB slices per respective rat were analyzed. Double positive cells were identified by comparing two color channels. The position of counted cells was saved as .roi files or .xml files. The number of cells was averaged per experimental rat and bulb. The counting was done in 4-6 non-overlapping sections of one olfactory bulb of a respective rat. The total number of positive cells was averaged over the number of counted sections, to account for variance in cell distribution and number of sections analyzed. In the OB double-labeled cells were counted, if the typical soma shape was visible without a staining of the nucleus area in the GFP-channel (green) and could be identified with the same outline in the pERK channel (magenta, see example in Fig. 1a). In the HDB double-labeled cells were counted, if the typical soma shape was visible in the pERK-channel (magenta) and could be identified with the same outline in the ChAT-channel (green, see example in Fig. 5a). For control the localization of the nucleus was double-checked in the DAPI channel (blue).

All the data was analyzed by an observer blinded with respect to stimulation groups.

### Electrophysiology

*Slice preparation*. 11-18 day-old juvenile VP-eGFP rats were used for in vitro electrophysiology experiments. The rats were deeply anesthetized with isoflurane and quickly decapitated. Horizontal and sagittal slices (300 µm) were cut in ice-cold carbogenized ACSF (artificial cerebrospinal fluid; mM: 125 NaCl, 26 NaHCO3, 1.25 NaH2PO4, 20 glucose, 2.5 KCl, 1 MgCl2, and 2 CaCl2) using a vibratome (VT 1200, LEICA) and afterwards incubated in ACSF at 36 °C for 45 min. Until experiments, the slices were kept at room temperature (~21 °C) in ACSF.

*Electrophysiology*. Brain slices were placed in a recording chamber on the microscope's stage perfused with carbogenized ASCF circulated by a perfusion pump (ISM 850, Cole-Parmer). GFP-labeled vasopressin cells (VPCs) were identified with LED illumination (470 nm) under a modified Zeiss Axioplan microscope (Carl Zeiss Microscopy, Oberkochen, Germany). Epifluorescence was filtered by a long-pass dichroic mirror (490 nm cutoff) and an emission filter (510 ± 21 nm) and visualized with a digital camera (VisiCAM-100, Visitron Systems, Puchheim, Germany). To perform whole-cell patch-clamp recordings, cells were visualized by infrared gradient-contrast illumination via IR filter (Hoya, Tokyo, Japan). Glass pipettes for recordings were pulled by a pipette puller (Narishige, Tokyo, Japan) sized 4-6 MΩ and filled with intracellular solution (in mM: 130 K-methylsulfate, 10 HEPES, 4 MgCl2, 4 Na2ATP, 0.4 NaGTP, 10 Na Phosphocreatine, 2 ascorbate, pH 7.2). Recordings were performed with the current-clamp configuration using an EPC-10 (HEKA, Lambrecht, Germany) digital oscilloscope. Series resistance was

measured 10-30 MΩ. The average resting membrane potential was −50 to −60 mV. Experiments were only started in case the patched cells had a holding current below approximately −50 pA and a stable resting membrane potential. Experiments were performed at room temperature (~21 °C).

In experiments with AOB stimulation, sagittal slices including the accessory olfactory bulb (AOB) were used. VPCs for these experiments were identified in the posterior-dorsal area of the MOB (Supplementary Fig. 2c) since intact projections of AOB mitral/tufted cells to this area in sagittal slices were demonstrated using antidromic electrical stimulation previously[24]. Glass pipettes, intracellular solution and settings of oscilloscope were same as experiments with horizontal slices.

**Electrical extracellular stimulation.** Olfactory nerve (ON) stimulation was performed with a glass pipette stimulation electrode sized around 2 MΩ. Glass pipettes were filled with ACSF. The electrode was connected to an external stimulator (STG 1004, Multi-Channel Systems, Reutlingen, Germany) controlled by a PC. The stimulation strength was adjusted with the stimulator's software (MC_Stimulus, V 2.1.5) and stimulation was triggered by the amplifier software (Patchmaster v2x73.5, HEKA). Stimulation pipettes were gently placed in the ON layer anterior to a cell to patch using a manual manipulator (LBM-7, Scientifica, East Sussex, UK) under optical control with the microscope. The stimulation strength was 50–400 µA for 100 µs. The stimulation was triggered every 30 s to avoid desensitization of the neural networks.

AOB stimulation was performed with a glass pipette stimulation electrode; the stimulator and the software were the same as for MOB ON stimulation. Stimulation pipettes were gently placed in the vomeronasal nerve layer or in the EPL/mitral cell layer in the AOB using a manual manipulator (LBM-7, Scientifica, East Sussex, UK) under optical control with the microscope. The stimulation strength was 50–500 µA for 100 µs.

**Ca²⁺ imaging.** Fluorescence was recorded by two-photon laser scanning microscopy on a Femto-2D microscope (Femtonics, Budapest, Hungary), equipped with a tunable, Verdi-pumped Ti:Sa laser (Chameleon Ultra I, Coherent, Glasgow, Scotland). The microscope was equipped with a 60x Nikon Fluor water-immersion objective (NA 1.0; Nikon Instruments, Melville, NY, USA), three detection channels (green fluorescence (epi and trans), red (epi) and infrared light (trans)), and controlled by MES v4.5.613 software (Femtonics).

VP-eGFP cells were identified in the green channel at an excitation wavelength of 950 nm. VPC bodies were patched in the whole-cell mode with patch pipettes filled with regular intracellular solution (see Electrophysiology), Alexa Fluor 549 (50 µM, Invitrogen) and the Ca²⁺ indicator OGB-1 (100 µM, Invitrogen, Thermo Fisher Scientific) were added for neurite visualization and Ca²⁺ imaging. Fluorescence transients and image stacks were acquired at 800 nm laser excitation. Data were mostly collected from the medial surface of the OB. Ca²⁺ imaging experiments were performed at room temperature (~21 °C). The patched VPCs were held in the current clamp mode near their resting potential of −55 mV. Structures of interest were imaged in free line-scanning mode with a temporal resolution of ~1 ms.

**Experimental design and data analysis.** All drugs (see Pharmacology) diluted in ASCF were bath-applied via the perfusion system. Recordings under pharmacology were performed at least 5 min after the onset of administration to ensure that the drugs reached the recorded cell. Two average traces from 3-5 recordings in each condition were analyzed. The data was averaged per condition. The amplitudes of PSPs were measured using Origin 2018b (Origin Lab Corporation, Northampton, MA, USA). The amplitudes of IPSPs following pharmacology were normalized to the amplitudes during the ACSF condition (100%). The amplitudes of EPSPs following bicuculline application with pharmacology were normalized to the amplitudes during the bicuculline condition (100%). APs, EPSPs, IPSPs in experiments with ACh application were defined as: APs, the existence of APs after ON stimulation; EPSPs, observation of depolarization with amplitudes of >1.5 mV; IPSPs, the absence of APs and EPSPs. In some experiments with ACh, current (40, 60, 80, 100 pA, 600 ms) was injected via the recording electrode into the cell to test if ACh application alters the intrinsic excitability in VPCs. Spiking rates of AP trains were calculated as the number of spikes divided by the duration of AP trains. Latency of the first spike was measured as the duration between the current injection onset and the peak of the first spike.

In some experiments, simultaneous to the electrophysiological recordings, intracellular Ca²⁺ transients were measured at the tuft and at the apical dendrite directly apical to the soma (see Ca²⁺ Imaging) following ON stimulation. Three consecutive focal line-scans were performed during ON stimulation (see Olfactory nerve stimulation). To confirm that the cell was correctly filled with Ca²⁺-dye, VPCs were also stimulated with somatic 50 Hz trains (20 APs) via the patch pipette which was shown to reliably trigger Ca²⁺ influx in case ON stimulation did not result in Ca²⁺ signals near the soma during olfactory nerve stimulation[8]. The three line scans per pharmacological condition were averaged for analysis. Dendritic Ca²⁺ transients were analyzed in terms of ΔF/F relative to the resting fluorescence F₀[71]. The time course of pharmacology and analysis were the same as for electrophysiological data mentioned above.

All the data was analyzed by an observer blinded with respect to pharmacology.

**Pharmacology.** All pharmacological agents used were diluted in ACSF for bath application: serotonin hydrochloride (20 µM, Sigma-Aldrich), DL-Norepinephrine hydrochloride (20 µM, Sigma-Aldrich), acetylcholine chloride (100 µM, Sigma-Aldrich), mecamylamine hydrochloride (20 µM, Sigma-Aldrich), atropine (10 µM, Sigma-Aldrich), (-)-Nicotine, Ditartrate (100 µM, Merck KGaA, Darmstadt, Germany), (+)-Muscarine chloride (1 µM, Sigma-Aldrich), 1(S),9(R)-(-)-bicuculline methyl bromide (50 µM, Sigma-Aldrich).

The doses used were chosen based on minimal doses reported to elicit in vitro effects[8,26,47,72–76].

**Behavioral pharmacology**

*Cannula implantation.* Three cohorts of 5–6 week-old male Wistar rats were group-housed until surgery. To allow local injection of pharmacological solutions in the OB without anesthesia, we bilaterally implanted guide cannulae (23 G, Injecta GmbH, Klingenthal, Germany) in the rat OB. Before surgery, all surgical instruments were autoclaved at 121 °C for 20 min. The rats were anesthetized by isoflurane using a TEC 3 isoflurane vaporizer (Eickemeyer, Tuttlingen, Germany). The concentration of isoflurane was maintained between 2 and 3%. The rats were fixed on a stereotaxic frame (TSE, Bad Homburg Germany/Kopf, Tujunga, CA, USA) using ear bars. To prevent hypothermia, the rats were kept warm using a warming pad (ThermoLux, Witte + Sutor, Murrhardt, Germany). Analgetics (buprenorphine, 0.1 mg/kg s.c., Bayer, Beline, Germany) and antibiotics (enrofloxacin, 10 mg/kg s.c., Bayer) were injected before surgery. After shaving the head and disinfection of the operation site with 70 % ethanol, we incised the scalp and the supporting tissue on the skull. Injection of lidocaine hydrochloride (2 %, around 100 µL, bela-pharm, Vechta, Germany) under the scalp was operated if needed. Two stainless steel jeweler's screws were inserted into the skull using a surgical drill (NM 3000, NOUVAG, Goldach, Switzerland). One on the left anterior to bregma and the other one on the right posterior to bregma to secure dental cement fixation. Holes for guide cannulae were made by a drill, followed by placement of cannulae 7 mm anterior and 1.2 mm lateral relative to bregma as well as 1 mm ventral from the surface of the skull where 2 mm above the injection coordinates. Thus, the coordinates for injection were 3 mm ventral to the surface of the skull[70]. Cannulae and the two screws were fixed together by the dental cement (Kallocryl, Speiko-Dr. Speier GmbH, Muenster, Germany). Once the cement dried, the connection between the cement and the scalp was disinfected by iodine solution and cannulae were blocked by insertion of stainless steel stylets (25 G, BD, Heidelberg, Germany) to prevent dust entering the cannulae that could result in infection or blockage. The rats were removed from the frame and placed in fresh cages for recovering from anesthesia under observation until they are awake. The rats were weighed prior to surgery and on the next day to see if they recovered properly after surgery. If needed, further analgaetic treatment was performed. Their health condition and behavior were checked for at least five days. After surgery, they were single-housed until experiments to prevent damage to the guide cannulae. Meanwhile, the rats were handled and habituated to the removal and insertion of stylets every day until experiments. Stylets were cleaned with 70 % ethanol every time they were removed from the guide cannula.

**Social discrimination with pharmacology.** We combined the social discrimination paradigm[4] and microinjection of pharmacology into the OB. The social discrimination paradigm consists of a sampling phase and a discrimination phase. Experiments were performed in the afternoon. In the sampling phase (4 min), one stimulus rat was gently introduced into the cage of the experimental rats. Then the stimulus rat was removed from the cage and the experimental rat stayed in the cage alone for an inter-exposure interval of 30 min. In the discrimination phase (4 min), two stimulus rats, one is the same as during the sampling phase and the other one is a novel stimulus rat, were gently introduced into the cage. During both phases, the behavior of the experimental rat was video-taped from above through a transparent plastic plate for post-hoc analysis. Stimuli were 3–4 week-old group-housed male rats that were single-housed only in between the social exposures of the behavioral experiments to prevent mixing of body odors. They were marked with a red or black pen of the same brand (Edding, Ahrensburg, Germany) to allow visual differentiation by the observers. Pharmacological agents diluted in ACSF include atropine (1 µg in 1 µL) and [Arg8]-vasopressin acetate salt (1 ng in 1 µL, Sigma-Aldrich). 40 min before the sampling phase, either ACSF or atropine was injected into both OB hemispheres (1 µL each). The 10 min before the sampling phase, either ACSF or VP was injected into both OB hemispheres (1 µL each). All injections were carried out with microinjection syringes (Hamilton, Bonaduz, Switzerland) connected to a self-made injection system which consists of plastic tubes and 10 mm-long 30 G needles (2 mm longer than the guide cannulae). After every injection (1 µL), the injection systems rested in place for 1 min to allow the injected solution to diffuse fully in the tissue. The doses used were chosen based on minimal does reported to elicit behavioral effects[6,77]. No anatomical outliers in stereotaxic cannula implantation had to be excluded from behavioral analysis (Fig. 6c). However, one animal from the control group was removed, as it showed a high amount of unspecific sexual/mounting behavior towards the stimulus animals throughout the sampling as well as the discrimination phase.

To confirm that neither atropine nor VP microinjection into the OB induced unspecific behavioral effects, we demonstrated that these manipulations did not interfere with non-social investigatory/play behavior and habituation (towards

**Table 1 Statistical overview.**

| | Figure | Type of statistic | Type of test | Test statistic | P value | Effect size | Power |
|---|---|---|---|---|---|---|---|
| a | S1a | Parametric | One-way ANOVA | $F_{(2,26)}=0.406$ | $p=0.670$ | $f=0.176$ | 0.115 |
| b | 1c | Parametric | One-way ANOVA and post-hoc (LSD) | $F_{(2,26)}=3.713$ | $p=0.038$ | $f=0.463$ | 0.548 |
| | | | Rat vs water | – | $p=0.011$ | $d=1.147$ | – |
| | | | Rat vs urine | – | $p=0.197$ | $d=0.604$ | |
| | | | Urine vs. water | – | $p=0.195$ | $d=0.663$ | – |
| c | 1e | Parametric | One-way ANOVA and post-hoc (LSD) | $F_{(2,26)}=3.586$ | $p=0.042$ | $f=0.525$ | 0.663 |
| | | | Rat vs water | – | $p=0.025$ | $d=1.117$ | – |
| | | | Rat vs urine | – | $p=0.034$ | $d=0.824$ | |
| | | | Urine vs. water | – | $p=0.949$ | $d=0.232$ | – |
| d | S1c | Parametric | One-way ANOVA | $F_{(2,24)}=0.633$ | $p=0.539$ | $f=0.229$ | 0.156 |
| e | S1d | Parametric | One-way ANOVA | $F_{(2,24)}=0.746$ | $p=0.485$ | $f=0.237$ | 0.164 |
| f | S1e | Parametric | One-way ANOVA | $F_{(2,24)}=1.952$ | $p=0.164$ | $f=0.403$ | 0.404 |
| g | 2c | Non-parametric | Related samples Wilcoxon test | $W_{(8)}=0; z=-2.521$ | $p=0.012$ | $r=0.891$ | 0.995 |
| h | 2e | Non-parametric | Related samples Wilcoxon test | $W_{(9)}=1; z=-2.547$ | $p=0.011$ | $r=0.849$ | 0.984 |
| i | 2 g | Non- parametric | Related samples Wilcoxon test | $W_{(20)}=0; z=-3.920$ | $p<0.001$ | $r=0.876$ | 1.000 |
| j | 2 h | Non-parametric | Mann–Whitney-U test | $z=-4.652$ | $p<0.001$ | $r=0.736$ | 1.000 |
| k | 2i | Non-parametric | Kruskal–Wallis and post-hoc (Bonferroni) | H = 17.439, df=2 | $p<0.001$ | – | – |
| | | | 5-HT vs NA | $z=0.003$ | $p=1.000$ | $r=0.001$ | 0.050 |
| | | | ACh vs 5-HAT | $z=-3.433$ | $p=0.003$ | $r=0.649$ | 0.991 |
| | | | ACh vs NA | $z=-3.291$ | $p=0.002$ | $r=0.611$ | 0.980 |
| l | S2a | Parametric | (4) × (2) mixed model ANOVA (intensity [within-subject] × treatment [within-subject]) | $F_{(3, 54)}=23.12$ | $p<0.001$ | $f=1.132$ | 1.000 |
| | | | | $F_{(1, 18)}=0.647$ | $p=0.432$ | $f=0.176$ | 0.119 |
| | | | | $F_{(3,54)}=0.562$ (interaction) | $p=0.604$ | $f=0.255$ | 0.158 |
| m | S2b | Parametric | (4) × (2) mixed model ANOVA (intensity [within-subject] × treatment [within-subject]) | $F_{(3, 54)}=52.19$ | $p<0.001$ | $f=1.705$ | 1.000 |
| | | | | $F_{(1, 18)}=3.228$ | $p=0.089$ | $f=0.423$ | 0.398 |
| | | | | $F_{(3,54)}=6.346$ (interaction) | $p=0.010$ | $f=0.594$ | 0.792 |
| | | | ACh vs. ACSF with 40 pA | – | $p=0.026$ | $d=1.083$ | – |
| n | 3b | Non-parametric | Related samples Wilcoxon test | $W_{(5)}=0; z=-2.023$ | $p=0.043$ | $r=0.905$ | 0.872 |
| o | 3d | Non-parametric | Related samples Wilcoxon test | $W_{(5)}=0; z=2.023$ | $p=0.043$ | $r=0.905$ | 0.872 |
| p | – | Non-parametric | Mann–Whitney-U test | $z=3.121$ | $p=0.002$ | $r=0.716$ | 0.988 |
| q | 3 g | Parametric | (2) × (2) mixed model ANOVA (location [within-subject] × treatment [within-subject]) | $F_{(1,7)}=17.8$ | $p=0.004$ | $f=1.596$ | 1.000 |
| | | | | $F_{(1,7)}=1.98$ | $p=0.203$ | $f=0.531$ | 1.000 |
| | | | | $F_{(1,7)}=0.014$ (interaction) | $p=0.910$ | $f=0.045$ | 0.081 |
| r | 3 g | Parametric | (2) × (2) mixed model ANOVA (stimulation [within-subject] × location [within-subject]) and post-hoc (Bonferroni) | $F_{(1, 7)}=4.194$ | $p=0.080$ | $f=0.775$ | 0.424 |
| | | | | $F_{(1, 7)}=7.446$ | $p=0.029$ | $f=1.030$ | 0.650 |
| | | | | $F_{(1,7)}=15.01$ (interaction) | $p=0.006$ | $f=1.454$ | 0.911 |
| | | | ON/soma vs. 50 Hz/soma | – | $p=0.014$ | $dz=5.943$ | – |
| | | | ON/soma vs. ON/tuft | – | $p=0.002$ | $dz=6.643$ | – |
| s | S3c | Non-parametric | Mann–Whitney-U test | $z=0.936$ | $p=0.456$ | $r=0.209$ | 0.148 |
| t | 4b | Non-parametric | Related samples Wilcoxon test | $W_{(8)}=4; z=-0.420$ | $p=0.327$ | $r=0.148$ | 0.065 |
| u | 4d | Non-parametric | Friedman test | $\chi2=3.714$, df=2 | $p=0.156$ | – | – |
| v | 4 f | Non-parametric | Related samples Wilcoxon test | $W_{(6)}=0; z=-2.201$ | $p=0.028$ | $r=0.984$ | 1.000 |
| w | 4 h | Non-parametric | Friedman test and post-hoc (Dunn) | $\chi2=6.750$, df=2 | $p=0.034$ | – | – |
| | | | Mecamylamine/ACh vs Mecamylamine | $z=2.250$ | $p=0.024$ | $r=0.795$ | 0.868 |
| | | | Mecamylamine/ACh vs ACSF | $z=-2.250$ | $p=0.024$ | $r=0.795$ | 0.868 |
| | | | Mecamylamine vs ACSF | $z=0.000$ | $p=1.000$ | $r=0.000$ | 0.050 |
| x | 5b | Parametric | One-way ANOVA | $F_{(2,13)}=2.701$ | $p=0.104$ | $f=0.645$ | 0.526 |
| y | 5c | Parametric | One-way ANOVA and post-hoc (LSD) | $F_{(2,13)}=4.411$ | $p=0.034$ | $f=0.823$ | 0.748 |
| | | | Rat vs water | – | $p=0.113$ | $d=0.918$ | – |
| | | | Rat vs urine | – | $p=0.011$ | $d=1.826$ | – |
| | | | Urine vs. water | – | $p=0.255$ | $d=0.933$ | |
| z | 6d | Parametric | Paired samples $t$ test | $t_{(12)}=-2.411$ | $p=0.033$ | $dz=0.709$ | 0.651 |
| a1 | 6e | Parametric | Paired samples $t$ test | $t_{(13)}=-1.231$ | $p=0.240$ | $dz=0.202$ | 0.208 |
| b1 | 6 f | Parametric | Paired samples $t$ test | $t_{(13)}=-2.420$ | $p=0.031$ | $dz=0.647$ | 0.647 |
| c1 | 6 g | Non-parametric | Kruskal–Wallis Test | $H=0.915$, df=2 | $p=0.633$ | | |
| d1 | 6 h | Parametric | (4) × (3) mixed model ANOVA (time bin [within-subject] × treatment [between-subject]) | $F_{(3,114)}=51.02$ | $p<0.001$ | $f=1.161$ | 1.000 |
| | | | | $F_{(2,38)}=0.599$ | $p=0.355$ | $f=0.244$ | 0.301 |
| | | | | $F_{(6,114)}=1.509$ (interaction) | $p=0.234$ | $f=0.283$ | 0.427 |
| e1 | S4a | Non-parametric | Kruskal–Wallis Test | $H=0.341$, df=2 | $p=0.843$ | – | – |
| f1 | S4b | Parametric | (4) × (3) mixed model ANOVA (time bin [within-subject] × treatment [between-subject]) | $F_{(3,114)}=82.09$ | $p<0.001$ | $f=1.147$ | 1.000 |
| | | | | $F_{(2,38)}=0.599$ | $p=0.554$ | $f=0.179$ | 0.142 |
| | | | | $F_{(6,114)}=1.144$ (interaction) | $p=0.224$ | $f=0.274$ | 0.456 |

Type of statistic was determined using the Kolmogorov- Smirnov test in SPSS. Tests were performed using SPSS. Effect sizes for parametric statistics were determined using SPSS and G*Power (Cohen's $d$, $dz$, $f$). Effect sizes for non-parametric statistics were calculated from z-scores (Pearson's $r = |z|/\sqrt{n}$) [73]. Power (for $\alpha = 0.05$) was determined using G*Power.

amyl acetate (Sigma-Aldrich)/ (-)-carvone (Sigma-Aldrich) presented in a teaball or a juvenile rat). The play behavior was measured during the sampling phase of the social discrimination experiment. The non-social investigatory behavior in a separate experiment on the next day. Experimental rats and treatments were the same as in the social discrimination experiment.

**Confirmation of injection sites**. Immediately after the last experiments, the rats were killed with $CO_2$. Then 1 µl of blue ink was injected with an injection system via the guide cannulae into both OBs followed by decapitation. The OB was extracted and quickly frozen in isobutanol on dry ice and stored at −20 °C until use. Correct placement of Injection was identified on 40 µm cryostat sagittal

sections stained with cresyl violet (Fig. 5c). No anatomical outliers in stereotaxic cannula implantation had to be excluded from behavioral analysis.

**Video analysis: social investigation**. The analysis was done by an observer blind to the pharmacological treatment using JWatcher (downloaded from https://www.jwatcher.ucla.edu/). Thereby the duration of investigation of the stimulus rats by the experimental rats was measured. Investigation was defined as sniffing the anogenital or neck region of stimulus rats including obvious nose movements (sniffing). Aggressive behavior (e.g., aggressive grooming), only staying next to the stimulus rats, and rough and tumble play were not considered investigation[4] (Fig. 5b). One animal from the control group was removed, as it showed high

amounts of unspecific sexual/mounting behavior towards the stimulus animals throughout the sampling as well as the discrimination phase.

**Statistics and reproducibility**. Statistics were performed with SPSS (ver. 26, IBM, Armonk, NY, USA) and G*Power (ver. 3.1.9.2, Franz Faul, University of Kiel). All statistical analysis performed was two-sided and significance was accepted at $p < 0.05$. Type of statistic was determined using the Kolmogorov-Smirnov test for normal distribution in SPSS. Effect sizes for parametric statistics (Cohen's $d$, $dz$, $f$) were determined using SPSS and G*Power. Effect sizes for non-parametric statistics were calculated from z-scores (Pearson's $r = |z|/\sqrt{n}$)[78]. Sample sizes were either based on prior studies or an a priori power analysis. Power and sample sizes (for $\alpha = 0.05$) were determined using G*Power.

All Ns indicate biological replication, that is, data from different samples (different cells or different animals). To better illustrate replication in acute in vitro slice experiments it is also clearly indicated in the results section from how many different animals the measurements originate from.

For complete details concerning type of statistic (parametric or non-parametric), type of test, test statistics (F, t, W, U, z-score), degrees of freedom, $p$ value, effect size, and achieved power of every statistical analysis performed in the result section see Table 1. Figure identifiers and lower case arabic letters in the table represent affiliation to the statistical data represented in the figures and the text, respectively.

**Reporting summary**. Further information on research design is available in the Nature Research Reporting Summary linked to this article.

## Data availability

All numerical data that is represented in the figures (Figs. 1–6, Supplements) available as source data file corresponding to the figures (Supplementary Data 1). All other data are available from the corresponding author (or other sources, as applicable) on reasonable request.

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

## Acknowledgements

The work was supported by the German research foundation (DFG LU2164/1-1 and EG 135/5-1). We thank Anne Pietryga-Krieger and Dr. Max Müller for experimental support and Dr. Mike Ludwig (University of Edinburgh) for providing the VP-eGFP rats. We also thank Prof. Inga Neumann for providing video equipment and stereotactic instruments for the behavioral pharmacology experiments and Dr. Vinicius Oliveira for critically reading the manuscript.

## Author contributions

H.S. and M.L. designed research; M.L. and H.S. performed research; M.L. and H.S. analyzed data; V.E. provided advice on the design and analysis of electrophysiology and Ca$^{2+}$ imaging experiments; H.S. wrote the first draft of the manuscript. H.S., V.E., and M.L. wrote and revised the manuscript.

## Funding

## Competing interests

The authors declare no competing interest.
