## [Peer Review File · Communications Biology]

Reviewers' comments:

Reviewer #1 (Remarks to the Author):

The study by Suyama et al. uses in vitro slice electrophysiological recordings and in vivo stimuli exposure and pharmacology in rats to shed light on the mechanisms by which bulbar vasopressin cells (VPC) are engaged in social discrimination. Using pERK as a marker for neuronal activity the authors found that bulbar VPCs are predominantly activated by stimulation with a conspecific. Using the same method cholinergic cells in the horizontal band of Broca were found to be activated during social interaction. In vitro, electrophysiology/pharmacology revealed the influence of muscarinic neuromodulation on olfactory nerve-evoked action potential generation in VPC. In vivo pharmacology finally showed an influence of cholinergic modulation on vasopressin-dependent social discrimination. Special care was taken to test for possible contributions of the accessory olfactory system.

Overall, this is an interesting study that highlights new potential mechanisms involved in social discrimination. The methods used, are appropriate, and carefully executed, however, like any methods, are not without limitations. While this is an elegantly written manuscript and this reviewer especially appreciates that a junior PI acts as a senior author, some of the claims in the manuscripts seem to be overstated and not justified by the data provided. E.g. the title "Top-down acetylcholine enables social discrimination via unlocking action potential generation in olfactory bulb vasopressin cells" and the abstract "In vitro slice electrophysiology combined with pharmacology and immunohistochemistry then demonstrated that centrifugal cholinergic inputs from the diagonal band of Broca can enable olfactory nerve-evoked action potentials in VPCs via muscarinic neuromodulation" is misleading since no direct evidence is given that e.g. top-down acetylcholine from the HDB unlocks action potentials in OB VPC cells. While this reviewer follows the authors' line of argumentation and agrees that this is a likely scenario, it would be good if the authors could use a bit more caution to not sell "interpretations" as "proven by the experiment".

Major Concerns

1) One of the most puzzling results is the difference between VPC activation in the urine alone and conspecific condition. While the authors are a bit more careful in choosing their wordings here (e.g. "multisensory social interaction condition"), the fact remains that urine alone and the conspecific condition are very dissimilar and a plethora of factors could contribute to the differences observed. E.g. can the authors be sure that it is social interaction with a juvenile rat that is causing the effects? Would they expect to see different results if the experiments would be repeated with an adult conspecific (as a control)? How about avoiding contact by separating rats with Plexiglas (is a visual stimulus sufficient; what happens if a visual stimulus is paired with urine similar to what the authors discuss for oxytocin in the Discussion; line 471). I refuse to be the reviewer who asks for these experiments but I believe that the data should be interpreted more carefully. Performing Ca²⁺ imaging of VPC during different conditions would be an elegant way of testing these hypotheses.

2) pERK HDB labeling. This might be the weakest point of the paper. 1) IHC for Chat is not very convincing 2) in contrast to the in vitro slice electrophysiological recordings just one area (the HDB) was examined, potentially leading to confirmation bias. Both 5HT and NA decrease IPSP amplitudes and it would have been interesting to also examine raphe, LC for activity modulations, potentially revealing synergistic effects. Additionally, HDB recordings in awake animals during social interaction/urine could shed light on faster modulatory effects.

3) "Is centrifugal modulation of bulbar VPCs essential for VP-dependent social discrimination?" In vivo pharmacology has to be interpreted carefully and systemic effects cannot be excluded. The use optogenetic of pharmacogenetic experiments to specifically silence cholinergic HDB neurons would have been informative. Additionally: was a Veh/VP group tested? What would be the prediction here?

Other Points

1) The authors classify VPC as non-bursting superficial tufted cells. Are VPC axonless? If not where do they project (Fig 6)?

2) E.g. Line 103 instead of stating the number of animals, the number of labeled cells would be more informative. Additionally, it would be helpful to state the statistical test used in the main text.

- 3) Can AOB inputs elicit excitation in MOB VPCs. Is there a way to verify that the connections from AOB MTCs to the MOB were still intact in the slice?
- 4) Line 203 ACh application alone. Is "alone" correct here or was ACh always applied in combination with bicuculline? Additionally, values for both conditions (bicu and ACh) should be stated in the text.
- 5) "ACh did not further increase Ca²⁺ influx neither in the tufts or the soma". Is that a real finding or a technical problem? Could a positive control be performed e.g. can Ca²⁺ increase be detected following current injection protocols?. The authors mention that they performed such measurements in the Methods. An example recording could be added to the figure as a comparison.
- 6) Most ACh blockers have diverse side effects or are not specific. This should be discussed.
- 7) Line 235 "These results indicate that the muscarinic pathway is the major player in both increasing excitation and disinhibition of ON-stimulated VPCs". Couldn't the results also be explained with an increased excitation without (GABAergic) disinhibition?
- 8) Line 273. Does the test used here differ from that in Fig1?. Why is just one p-value mentioned? Was a multiple comparison test used?
- 9) Line 364 "Thus, we cannot exclude that non-specific activity-induced pERK possibly masks stimulus-specific activation of AOB M/TCs." If this is the case the amount of baseline pERK cells should be higher compared to other brain areas. Was this indeed the case?
- 10) Line 459 "Thus, muscarinic neuromodulation in the OB is more important for facilitating the memory of social odor signatures." It is not clear what the authors mean by facilitating the memory of social odor signatures.
- 11) Same line "These findings indicate that social discrimination/habituation and simple odor discrimination/habituation paradigms differ from each other". Or the approaches/paradigms used to study them?
- 12) Line 477 the appreciation (V1aR) should be introduced.
- 13) The AOB stimulation seems to be missing in the Methods
- 14) Cannula coordinates seem to be contradictive: Line 714 2mm above injection side, Line 715 3 mm ventral to the surface, Line 729 1 mm ventral from the surface. Stylet placement/removal should be described in more detail.
- 15) Could the vendor and part number for the Stylets be added?
- 16) Does Fig 2G and 3B show the same data?

Reviewer #2 (Remarks to the Author):

In this manuscript, Suyama and colleagues examined the activation of bulbar vasopressin cells (VPCs) during social interactions in rats. The main goal of the study was to elucidate the synaptic mechanisms and centrifugal neuromodulators that are involved in VPCs activation during social discrimination. The authors based their working hypothesis on their previous study which showed that VPCs receive excitatory inputs from the olfactory nerve (ON) and electric ON stimulation did not excite VPCs, but induced IPSPs. The authors then raised the question of how intrinsic VP neurotransmission is possible during social interactions if olfactory inputs inhibit VPCs. The authors combined slice electrophysiology, behavior and pharmacology to demonstrate that top-down cholinergic modulation of bulbar VPC activity via muscarinic receptors is a key player for social discrimination in rats. The manuscript is quite interesting, however there are some concerns with their results and how they are interpreted that need to be addressed.

In Figure 1, the authors show quantifications of the number of VPCs, pERK+ VPCs and pERK+ MCs. How was the quantification done? Some more details are needed in the methods paragraph in order for the reader to understand the stainings. Did the authors normalize the number of cells to the area? The authors should consider to present the normalized density of the cells between brain sections and animals. In addition, it is recommended to split the channels (green and magenda) in addition to showing the composite. This can help the reader visualize better the cells that are colocalized with GFP and pERK.

In Figure 3A, the authors observed somatic inhibition upon ON stimulation. When they bath applied ACh, inhibition was reduced (Fig. 3B). Yet, they don't see Ca²⁺ signals being propagated to the soma (Fig. 3F). But, during ACh application, inhibition was reduced (A and B), so there should have been a change in somatic Ca²⁺ with calcium imaging. So, based on their findings of

Fig. 3E-G, GABAergic inhibition does not account for the lack of somatic Ca²⁺ signaling, nor does ACh. In the text, the authors claim "The finding of a distal postsynaptic Ca²⁺ entry that cannot be detected near the soma, is consistent with the concurrent observation of strong GABAergic inhibition at the soma suggesting that excitation at the tufts is inhibited alongside the apical dendrite". This statement needs to be supported either experimentally or the authors should provide proper references.

In Figure 4, the authors performed immunostainings of the neurons in the diagonal band of Broca against pERK and choline acetyltransferase using the brains from the exposure experiments. They came to the conclusion that cholinergic cells in the HDB are activated during social interaction, however, they do not find any significant difference between the number of pERK+ cholinergic cells comparing between water and conspecific (Fig.4C). This should be explained in the text. In Supplemental Figure 1, the authors should consider splitting the two channels (green and magenta). The colocalization of the cells is not evident. Also, a panel with higher magnification images could help the reader appreciate the work.

Finally, in Supplemental Figure 2D the authors claim that electrical AOB stimulation does not evoke excitatory responses in MOB VPCs. The authors should show the representative averaged traces of responses instead of a single trace.

Reviewer #3 (Remarks to the Author):

In this paper, the authors demonstrate the existence of a top-down regulation from the basal forebrain cholinergic system that, via VPC neurons, is involved in social discrimination. The authors show that activation of bulbar VPC neurons requires the additional activation of the basal forebrain cholinergic input acting via muscarinic and nicotinic receptors. While the experiments are generally straightforward, the conclusions are not. Overall, a lack of mechanistic explanations for the effects dampen my enthusiasm for the paper. Statistical approaches are adequate.

Major Concerns

1) If pERK is taken as a marker for neuronal activation, based on Fig. 1, it appears that the entire neuronal population of the bulb show activation, not just VPC neurons, upon social exposure. This suggests general arousal mechanisms rather than a specific modulation of VPC neurons. This might explain why lesser number of VPC neurons are activated with urine, that would only activate a subset of the glomerular circuits. Would the authors predict similar increase in VPC activation in all perceptual learning tasks, e.g., discrimination of closely related odors, that also involves the cholinergic inputs? The point here is that the results might be true for all behaviors involving attention and learning rather than a specific example of multisensory integration in the social paradigm used in this paper.

2) Bath application of agonists is not a good way to elicit responses. Firstly, it activates receptors across the bulb making it difficult to identify specific targets. Secondly, it is not an effective means for activating receptors that can desensitize like the nicotinic receptors, which might explain partial effects seen with nicotinic blockade. In fact, the idea that nicotinic receptors might be involved in gain control mechanisms (J.Neurosci 32,3261,2012) similar to that discussed in the paper makes this an important point to address.

3) Coupled with the application method used, the pharmacology needs a more thorough elucidation. Mecamylamine at 20 μ M is unlikely to block the $\alpha 7$ nicotinic receptors, which is densely expressed in the glomerulus. Similarly, atropine, at 10000-fold the K_i for muscarinic receptors (why use this high a concentration?) can also affect nicotinic currents. How did the authors conclude that the relevant receptors were completely blocked and that other types were not affected? At least two chemically distinct antagonists should be used to confirm receptor specificity. Also, the authors might use selective agonists (nicotine and muscarine, not oxotremorine that can have nicotinic effects).

4) Based on the above two points, relative contributions of the two AChR subtypes cannot be estimated at this time.

5) In the calcium imaging experiments, the authors should do the experiments where the calcium concentration is initially clamped at about 50 nM using a mixture of calcium chloride and BAPTA or EGTA. The concern is that certain components of calcium signaling, e.g., receptor-coupled ER store release might not be operational under the conditions used.

Addressing these issues would significantly improve this paper.

In particular, please note that the following revisions would be necessary for us to contact our referees again:

(1) Provide higher-quality IF images, particularly for the ChAT staining, as noted by Referees #1-2.

Reply: We now provide higher quality IF images of the IHC stainings (higher contrast by computational clearing and higher magnification) in both, Fig. 1a+b (pERK/GFP) and Fig. 5a (pERK/ChAT). The higher quality representative images illustrate that it is possible to identify labelled cells individually in the GFP, pERK, and ChAT stainings. Following the suggestion of referee #2, we also split the green and magenta channels in Fig 1a and Suppl. Fig 1 and added an insert panel with higher magnification of cells of interest to the Fig.5a.

(2) Provide additional evidence for VPC-specificity, potentially through the optogenetic inhibition experiment suggested by Referee #1 and/or muscarinic receptor pharmacological experiments suggested by Referee #3.

Reply: We have now performed additional analysis of behavioral data to support the VPC-specificity of the behavioral effects during our pharmacological in-vivo experiments regarding possible unspecific systemic effects of the pharmacological agents. We further performed additional in-vitro experiments concerning the muscarinic pharmacology that yield very clear results, which in our view lessens the need for optogenetic silencing experiments. In the answer to the single comments, we also give a detailed explanation why the optogenetic inhibition experiments suggested by referee #1 would be very hard to perform and limited in their usefulness.

However, we are aware of the methodological limitations of our approach and we therefore, as the referee suggests, interpret our results more carefully via changing our statements throughout the manuscript to “Top-down cholinergic modulation of bulbar VPC activity is involved [instead of crucial/essential] in individual vasopressin-dependent social discrimination in rats” (abstract, l. 504).

(3) Justify the current experimental conditions or repeat calcium imaging under the conditions suggested by Referee #2.

Reply:

While we agree with the reviewer that the recording conditions are not ideally suited for a detailed investigation of Ca²⁺ signaling pathways, in the current study we used Ca imaging mainly as a readout for local depolarization. Moreover, with the very same internal solution (100 μM OGB-1), resting concentrations of ~ 50 nM were detected using the Maravall et al. (2000) method in granule cells, mitral cells and hippocampal pyramidal cells (Egger & Stroh 2009, Rammes et al., 2015), and release from ER stores was demonstrated to substantially contribute to intracellular postsynaptic Ca²⁺ signaling in granule cell spines (Egger et al., 2005). Thus, we would assume that Ca²⁺ signaling would not be overly disrupted. Also, since we have shown that 50 Hz-AP trains can elicit Ca²⁺ signals in the soma and the proximal apical dendrites, the lack of such signals in response to single APs is not a technical artefact but probably due to a very low density of voltage-gated Ca²⁺ channels in these compartments (Fig. S3b; ON/soma vs. 50Hz/soma p=0.014; n=8 from 8 rats), as compared to e.g., granule cell or mitral cell dendrites in which Ca²⁺ responses to single APs could easily be detected (e.g. Egger & Stroh, 2009). In any case, in the imaging experiments (with 100 μM OGB-1 in the cell) there was no difference in the

effect of ACh on the increase of ON-evoked PSP compared to pure electrophysiological experiments (Fig. S3c, $p=0.456$; $n=11$ vs. 9) and therefore this crucial effect is independent of the potential buffering of postsynaptic Ca^{2+} by the Ca^{2+} indicator.

Egger, V., Svoboda, K., and Mainen, Z.F. (2005). Dendrodendritic synaptic signals in olfactory bulb granule cells: Local spine boost and global low-threshold spike. *J Neurosci* 25, 3521-3530.

Egger, V., and Stroh, O. (2009). Calcium buffering in rodent olfactory bulb granule cells and mitral cells. *J Physiol* 587, 4467-4479.

Maravall, M., Mainen, Z.F., Sabatini, B.L., and Svoboda, K. (2000). Estimating Intracellular Calcium Concentrations and Buffering without Wavelength Ratioing. *Biophysical Journal* 78, 2655-2667.

Rammes, G., Gravius, A., Ruitenber, M., Wegener, N., Chambon, C., Sroka-Saidi, K., Jeggo, R., Staniaszek, L., Spanswick, D., O'hare, E., Palmer, P., Kim, E.-M., Bywalez, W., Egger, V., and Parsons, C.G. (2015). MRZ-99030 – A novel modulator of A β aggregation: II – Reversal of A β oligomer-induced deficits in long-term potentiation (LTP) and cognitive performance in rats and mice. *Neuropharmacology* 92, 170-182.

(4) Qualify the Discussion and expand the Methods, as suggested by all referees.

Reply: We addressed all the comments and followed the suggestions by all of the three referees concerning the discussion and methods section. For details, please see replies to single referee comments.

New Title: "Top-down acetylcholine contributes to social discrimination via unlocking action potentials in olfactory bulb vasopressin cells"

Major Concerns

1) One of the most puzzling results is the difference between VPC activation in the urine alone and conspecific condition. While the authors are a bit more careful in choosing their wordings here (e.g. "multisensory social interaction condition"), the fact remains that urine alone and the conspecific condition are very dissimilar and a plethora of factors could contribute to the differences observed. E.g. can the authors be sure that it is social interaction with a juvenile rat that is causing the effects? Would they expect to see different results if the experiments would be repeated with an adult conspecific (as a control)? How about avoiding contact by separating rats with Plexiglas (is a visual stimulus sufficient; what happens if a visual stimulus is paired with urine similar to what the authors discuss for oxytocin in the Discussion; line 471). I refuse to be the reviewer who asks for these experiments but I believe that the data should be interpreted more carefully.

Performing Ca^{2+} imaging of VPC during different conditions would be an elegant way of testing these hypotheses.

Reply: The main aim of the social and urine stimulation experiment was to investigate if and under which conditions olfactory bulb vasopressin cells are activated in-vivo, in contrast to the artificial in-vitro slice condition where VPCs were inhibited by olfactory nerve stimulation. The referee is right that it is not very clear which factors are responsible for the different levels of VPC activity between the groups. However, since only in the rat exposure group the amount of activated vasopressin cells in the bulb was

significantly higher than control, this experiment confirmed our hypothesis that neuronal processes beyond olfactory input (probably via top-down modulation) play a role in activating VPCs. E.g., influences like multimodal sensory stimulation, arousal or the need of attention towards the conspecific. Obviously, rat urine and even control water stimulation can activate vasopressin cells, even though rat stimulation was the most effective. This may be a result of weak arousal, tactile and/or pheromonal inputs from the water control and the urine. In addition, certain background activity of VPCs *in-vivo* in comparison to the constant inactivation *in-vitro* can be expected as it is not possible, under our laboratory conditions; to deprive experimental rats completely from all sensory inputs from outside their cage.

To address the concern of the referee, we revised the results part and the discussion adding explanations (see above) and tried to remove potential overinterpretation concerning the difference between urine and rat exposure that are not clearly supported by our experimental data (ll. 90-91, ll. 99-117, ll. 129-132, ll. 367-374, ll. 483-488, ll. 533-541).

To the other question of the referee. If the rat is juvenile or adult, is not that important for us in this context. The reason we chose a juvenile is that we wanted to mimic the stimulation that happens during a social discrimination test (which is classically done with juveniles or urine, Engelmann et al., *Nature protocols*, 2011). Accordingly, the used rat urine samples were also from juvenile rats that were similar age as the conspecific stimuli. However, we expect a similar high amount of activation, if we would use an adult, with slight variations depending how arousing the adult is to the tested rat (aggressive male, receptive female, neutral). The referee is right that the exact ethological/sensory/physiological identity of the factor that may be responsible for the higher activation could be found by additional testing. E.g., via separating the stimuli provided by a conspecific and potentially using a more temporal precise activity detection method like *in-vivo* Ca²⁺ imaging, but we think as the referee indicated himself that this would be too ambitious to be included in this data set.

Engelmann, M., Hädicke, J., and Noack, J. (2011). Testing declarative memory in laboratory rats and mice using the nonconditioned social discrimination procedure. *Nature Protocols* 6, 1152-1162.

2) pERK HDB labeling. This might be the weakest point of the paper.

2.1) IHC for Chat is not very convincing

Reply: We now provided higher quality IF images of the IHC stainings (higher contrast by computational clearing and higher magnification) in both, Fig. 1a+b (pERK/GFP) and Fig. 5a (pERK/ChAT). The higher quality representative images illustrate that it is possible to identify labelled cells individually in the pERK, and ChAT staining, even though the ChAT staining shows some background. Following the suggestion of referee #2, we also split the green and magenta channels in Fig 1a and Suppl. Fig 1 and added an insert panel with higher magnification of cells of interest to the Fig.5a. Further, we also indicate now in the abstract that we only “suggest” that cholinergic modulation of bulbar VPCs originates from the HDB.

2.2) in contrast to the *in vitro* slice electrophysiological recordings just one area (the HDB) was examined, potentially leading to confirmation bias. Both 5HT and NA decrease IPSP amplitudes and it would have been interesting to also examine raphe, LC for activity modulations, potentially revealing synergistic effects. Additionally, HDB recordings in awake animals during social interaction/urine could shed light on faster modulatory effects.

Reply: The focus on the cholinergic system and thereby the decision to investigate specifically the activation of HDB cholinergic neurons following social interaction is indeed inspired by our result that cholinergic pharmacological application in our electrophysiology experiments resulted in the most pronounced excitatory effect in VPCs following olfactory nerve stimulation (obtaining EPSPs and even action potentials whereas the other two only decreased inhibition). Nevertheless, in accordance with the comment of the reviewer, we also think that it is very likely that not only HDB cholinergic centrifugal modulation is responsible for increased vasopressin cell excitability, but also that serotonergic and noradrenergic centrifugal modulation from the raphe and LC, respectively, can have synergistic effects. Unfortunately, we cannot provide further immunohistochemistry experiments in the other brain regions as the leftover brain tissue of the stimulation experiments is no longer usable for staining since they are too old by now. A repetition of these experiments would ask for a new ethical approval by the local authorities and we do not think that the gain in information would justify a new set of animal experiments, as the focus of the manuscript is to shed light on the prominent cholinergic-vasopressin interactions. The same argument applies to the suggested *in-vivo* electrophysiological experiments. However, we agree that we should have discussed the high likelihood of serotonergic and noradrenergic involvement in these processes. Therefore, we added a passage concerning this topic in the discussion section. "Although here we focused on the cholinergic system because of the strength of cholinergic effects, our *in-vitro* electrophysiology results indicate that increased VPC excitability could be triggered by synergistic cholinergic, serotonergic and noradrenergic modulation; the possibility of such cooperative action should be accounted for in future studies." (ll. 455-458)

3) "Is centrifugal modulation of bulbar VPCs essential for VP-dependent social discrimination?" In vivo pharmacology has to be interpreted carefully and systemic effects cannot be excluded. The use of optogenetic or pharmacogenetic experiments to specifically silence cholinergic HDB neurons would have been informative. Additionally: was a Veh/VP group tested? What would be the prediction here?

Reply: We agree with the referee that in-vivo pharmacological manipulation of behavior can never completely mimic the complicated neurobiological and physiological processes during naturally occurring neuromodulator release. However, we decided to choose this approach out of several conceptual and methodological reasons:

Endogenous intrinsic vasopressin release in the olfactory bulb is essential for social discrimination (Tobin et al. 2010, extensively demonstrated with pharmacology, siRNA, genetically-targeted cholera toxin depletion). Our main aims were (1) to demonstrate that bulbar acetylcholine release is also involved in social discrimination (other groups have so far only shown non-local cholinergic involvement with VAcT-KO-mice; Prado et al. 2006, Neuron) and (2) to test, if the modulatory excitatory effects of acetylcholine on VPC activity that we found in-vitro also play a role in vasopressin-dependent social discrimination. Our pharmacological experiment approaches these aims with the least invasive approach (1x 2 guide cannulas implantation, 2x painless infusion of ACh-Antagonist and vasopressin agonist). An optogenetic/pharmacogenetic approach could for sure also meet our aim (1), but for the investigation of our aim (2), with the same local specificity, this experiment would ask for the surgical infusion of 2 different viral constructs as well as the implantation of 2x2 optical fibers in the bulb and HDB, 2 more painful i.p. injections or a combination of both. Moreover, while we agree with the referee that a single optogenetic/pharmacogenetic manipulation would be a more physiological approach for reaching aim (1), there are some methodological obstacles: we are not aware that there is a commercially available viral vector expressing inhibitory channelrhodopsin or hM4Di that gets specifically expressed in cholinergic neurons. Thus, the alternative would be to change experimental animal species from rats to a cre-mouse model. Due to species-specific differences in social behavior, comparing behavioral and electrophysiological effects in different species are often problematic.

For all these reasons, we do not believe that the gain in further information for this experiment outweighs the effort of establishing the technically challenging optogenetic/pharmacogenetic silencing approaches and does not justify a new set of animal experiments from an animal ethics perspective. Especially, as the only known source of cholinergic neuromodulation in the bulb so far are HDB projections from the basal forebrain (Záborsky et al. 1986, Rothermel et al. 2014). As cited in the discussion (l. 482-484), it was shown that non-ACh specific optogenetic silencing of overall release from HDB projections in the OB impairs social habituation to female conspecific mice (Schwarz et al., 2021, bioRxiv).

To address the concern of the referee and editor that pharmacological infusion may result in unspecific, possibly systemic, effects on vasopressin-dependent social discrimination behavior, we performed further analyses (new supplemental Fig. 4) demonstrating that VP and atropine microinjections into the OB do not interfere non-social investigatory/play and social play behavior as well as normal habituation to odor stimulation and social interaction. If atropine indeed resulted in reduced arousal or impaired general olfactory processing via unspecific effects inside or outside the olfactory bulb, we would expect changes in those behaviors driven by arousal and olfaction. Further, the doses of atropine (2x1µg) and vasopressin (2x1 ng) that we used for the in-vivo bulbar infusion are very low. Vasopressin as a peptide

cannot pass the blood brain barrier in low concentrations. In case the barrier was destroyed due to the invasive surgery and infusion, we know that systemic infusions of vasopressin in rats only result in neuronal activation (Bold signal) in the bulb itself and the hindbrain, not in limbic areas, and then only in concentrations ranging from 0.1 to 2.5 mg/kg (Ferris et al. 2015). In addition, atropine, in case low amounts reach the bloodstream, was shown not to induce changes in rat social interactive behavior even after chronic systemic application of 1 mg/kg (Liebenberg et al. 2012).

We did not test a Veh/VP group on purpose since we know from literature (Dluzen et al. 1998) that bulbar infusion of vasopressin does not result in a more pronounced discrimination after 30 min, but a prolonged maintenance of this discrimination ability. Since we did not test after a longer inter-exposure interval and as this prolongation effect was not of interest in our study, we omitted this control group, especially considering the 3-R rule, which demands to reduce animal numbers in in-vivo experiments whenever possible.

However, we are aware of the methodological limitations of our approach and we therefore, as the referee suggests, interpret our results more carefully via changing our statements throughout the manuscript to “Top-down cholinergic modulation of bulbar VPC activity is involved [instead of crucial/essential] in individual vasopressin-dependent social discrimination in rats” (abstract, l. 504).

Dluzen, D.E., Muraoka, S., Engelmann, M., and Landgraf, R. (1998). The effects of infusion of arginine vasopressin, oxytocin, or their antagonists into the olfactory bulb upon social recognition responses in male rats. *Peptides* 19, 999-1005.

Ferris, C.F., Yee, J.R., Kenkel, W.M., Dumais, K.M., Moore, K., Veenema, A.H., Kulkarni, P., Perkybile, A.M., and Carter, C.S. (2015). Distinct BOLD Activation Profiles Following Central and Peripheral Oxytocin Administration in Awake Rats. *Frontiers in Behavioral Neuroscience* 9.

Liebenberg, N., Harvey, B.H., Brand, L., Wegener, G., and Brink, C.B. (2012). Chronic treatment with the phosphodiesterase type 5 inhibitors sildenafil and tadalafil display anxiolytic effects in Flinders Sensitive Line rats. *Metab Brain Dis* 27, 337-340.

Prado, V.F., Martins-Silva, C., De Castro, B.M., Lima, R.F., Barros, D.M., Amaral, E., Ramsey, A.J., Sotnikova, T.D., Ramirez, M.R., and Kim, H.-G. (2006). Mice deficient for the vesicular acetylcholine transporter are myasthenic and have deficits in object and social recognition. *Neuron* 51, 601-612.

Rothermel, M., Carey, R.M., Puche, A., Shipley, M.T., and Wachowiak, M. (2014). Cholinergic Inputs from Basal Forebrain Add an Excitatory Bias to Odor Coding in the Olfactory Bulb. *The Journal of Neuroscience* 34, 4654-4664.

Schwarz, I., Müller, M., Pavlova, I., Schweihoff, J., Musacchio, F., Mittag, M., Fuhrmann, M., and Schwarz, M.K. (2020). The diagonal band of broca continually regulates olfactory-mediated behaviors by modulating odor-evoked responses within the olfactory bulb. *bioRxiv*, 2020.2011.2007.372649.

Záborszky, L., Carlsen, J., Brashear, H.R., and Heimer, L. (1986). Cholinergic and GABAergic afferents to the olfactory bulb in the rat with special emphasis on the projection neurons in the nucleus of the horizontal limb of the diagonal band. *Journal of Comparative Neurology* 243, 488-509.

Other Points

1) The authors classify VPC as non-bursting superficial tufted cells. Are VPC axonless? If not where do they project (Fig 6)?

Reply: We investigated these matters in our previous publication where we also reconstructed axonal projections of VPCs (Lukas et al., 2019, eNeuro). Our study revealed that VPCs feature vertical fan-like axonal projections in the glomerular, external plexiform, and superficial granule cell layers. A subset of VPCs projects further alongside the internal plexiform layer, but the target of these projections is so far unknown. We added the information concerning the axonal projection to Fig. 7 (summary scheme) and the introduction. *“Recently, we classified these bulbar VPCs as non-bursting superficial tufted cells, featuring an apical dendritic tuft that ramifies within a glomerulus, lateral dendrites that are located within the top part of the EPL and extended axonal ramifications, mostly within the entire EPL”* (II. 67-69)

2) E.g. Line 103 instead of stating the number of animals, the number of labeled cells would be more informative. Additionally, it would be helpful to state the statistical test used in the main text.

Reply: We now state the absolute number of VPCs analyzed per stimulation group in addition to the animal number (I. 107-111) as well as the statistical test used (see also reply to point 8). In order to enhance the readability of the results section and comply with the word limit of the journal we combined the full information about statistical testing in the statistical table 1. The subscript letter at the end of the brackets containing the representative p-value indicates the respective statistical information in the table.

3) Can AOB inputs elicit excitation in MOB VPCs. Is there a way to verify that the connections from AOB MTCs to the MOB were still intact in the slice?

Reply: For now, we cannot verify in our own *in-vitro* setting whether potential connections from the AOB to the MOB were intact. However, we now explain in the methods section that we patched VPCs in the posterior-dorsal area of the MOB (near the AOB) since intact projections of AOB mitral/tufted cells to this area in sagittal slices were demonstrated using antidromic electrical stimulation (Vargas-Barroso et al., 2016, II. 722-727). It was not our intention to exclude that these connections may exist or that pheromonal/AOB signaling is involved in the excitation of VPCs, as we discuss the possibility of pheromones involved in VPC excitation during social interactions in the discussion (II. 391-395, II. 402-411). Further, we changed the former representation in Supplemental Fig. 2d for a representative response of averaged traces to improve the quality of the figure.

Vargas-Barroso, V., Ordaz-Sanchez, B., Pena-Ortega, F., and Larriva-Sahd, J.A. (2015). Electrophysiological Evidence for a Direct Link between the Main and Accessory Olfactory Bulbs in the Adult Rat. *Front Neurosci* 9, 518.

4) Line 203 ACh application alone. Is “alone” correct here or was ACh always applied in combination with bicuculline? Additionally, values for both conditions (bicu and ACh) should be stated in the text.

Reply: In this experiment, the AP onsets from ACh only experiments from Fig. 2g were compared with the EPSP onsets from bicuculline-alone (before ACh) experiments from Fig. 3c. To avoid confusion, we changed the section as follows. “In addition, we analyzed the onset of ON-evoked APs in VPCs triggered by ACh modulation from Fig. 2 (8.6 ± 0.7 ms; n=12 from 12 rats, corresponding to Fig. 2g) and compared them to the onset of evoked EPSPs in bicuculline alone, which was significantly slower (32.8 ± 12.1 ms; n=6 from 5 rats, corresponding to Fig. 3c; $p=0.002$), further supporting the idea of additional excitation by ACh.” (II. 213-217)

5) “ACh did not further increase Ca²⁺ influx neither in the tufts or the soma”. Is that a real finding or a technical problem? Could a positive control be performed e.g. can Ca²⁺ increase be detected following current injection protocols?. The authors mention that they performed such measurements in the Methods. An example recording could be added to the figure as a comparison.

Reply: As suggested by the referee we added the control experiment using 50Hz-AP trains evoked by somatic current injection to the results section comparing to ON-evoked Ca²⁺ changes (II. 228-232) and present the cumulative data and representative Ca²⁺ traces in Supplemental Fig. 3.

6) Most ACh blockers have diverse side effects or are not specific. This should be discussed.

Reply: This issue was also criticized by referee #3. To address the specificity issues, we performed additional experiments involving the selective agonists nicotine and muscarine. For details, please refer to the response to comments 2-4 of referee #3.

7) Line 235 “These results indicate that the muscarinic pathway is the major player in both increasing excitation and disinhibition of ON-stimulated VPCs”. Couldn’t the results also be explained with an increased excitation without (GABAergic) disinhibition?

Reply: We admit that our wording in this statement may be misleading. When we wrote disinhibition of ON-stimulated VPCs, we did not want to indicate that this has to be inhibition of inhibitory inputs (GABAergic), but rather a reduction of the inhibitory synaptic signal following ON-stimulated VPCs. Thus,

we changed the statement to: “Thus, these results indicate that the muscarinic pathway is the major player in reducing inhibition of ON-stimulated VPCs,...” (ll. 278-280)

8) Line 273. Does the test used here differ from that in Fig1?. Why is just one p-value mentioned? Was a multiple comparison test used?

Reply: For Fig. 4 (Now Fig. 5) the same statistical test was used as for Fig. 1 (One-way ANOVA followed by a LSD multicomparison test, if appropriate). The single p-value reported for Fig. 5b represents the significance level of the ANOVA. As the null hypothesis that the values are the same was not rejected, there was no reason to perform a post-hoc comparison. Therefore, only one p-value was reported in this case. To make this clearer we now always report the p-value for the ANOVA followed by the p-values of the multiple comparison, if the ANOVA was significant.

9) Line 364 “Thus, we cannot exclude that non-specific activity-induced pERK possibly masks stimulus-specific activation of AOB M/TCs.” If this is the case the amount of baseline pERK cells should be higher compared to other brain areas. Was this indeed the case?

Reply: We are afraid that we do not understand the point of the referee. In addition, we do not see how the baseline comparison in pERK activity of different cell types from different brain regions of a specific marker that is, like c-Fos, not expressed in every cell and every brain region, could be statistically useful. However, as our statement resulted in confusion, we omitted it from the discussion.

10) Line 459 “Thus, muscarinic neuromodulation in the OB is more important for facilitating the memory of social odor signatures.” It is not clear what the authors mean by facilitating the memory of social odor signatures.

Reply: We admit that our wording in this statement may be confusing. In the revised version of the manuscript, we now do not use this statement anymore (ll. 506-517).

11) Same line “These findings indicate that social discrimination/habituation and simple odor discrimination/habituation paradigms differ from each other”. Or the approaches/paradigms used to study them?

Reply: Again, our wording in this statement may be confusing. We changed the sentence accordingly: “These findings indicate that social discrimination/habituation paradigms and simple odor discrimination/habituation paradigms differ from each other.” (ll. 515-517)

12) Line 477 the appreciation (V1aR) should be introduced.

Reply: We changed this unnecessary abbreviation to V1a receptor knock-down.

13) The AOB stimulation seems to be missing in the Methods

Reply: We added the AOB stimulation in the methods section (ll. 740-744)

14) Cannula coordinates seem to be contradictive: Line 714 2mm above injection side, Line 715 3 mm

ventral to the surface, Line 729 1 mm ventral from the surface. Stylet placement/removal should be described in more detail.

Reply: Our final coordinate of injection was 3 mm ventral to the surface. However, to preserve brain tissues, we implanted guide cannulae at only 1 mm ventral from the surface. Thus, the coordinate of cannulae implantation was indeed 2 mm above the injection site. To avoid confusion, we described that more clearly now (ll. 823-826, l. 859).

15) Could the vendor and part number for the Stylets be added?

Reply: Added accordingly.

16) Does Fig 2G and 3B show the same data?

Reply: Yes, data in 3b are the 25 % IPSPs following ACh application. We now indicated this clearer in the text (ll.200-202).

Reviewer #2 (Remarks to the Author):

In this manuscript, Suyama and colleagues examined the activation of bulbar vasopressin cells (VPCs) during social interactions in rats. The main goal of the study was to elucidate the synaptic mechanisms and centrifugal neuromodulators that are involved in VPCs activation during social discrimination. The authors based their working hypothesis on their previous study which showed that VPCs receive excitatory inputs from the olfactory nerve (ON) and electric ON stimulation did not excite VPCs, but induced IPSPs. The authors then raised the question of how intrinsic VP neurotransmission is possible during social interactions if olfactory inputs inhibit VPCs. The authors combined slice electrophysiology, behavior and pharmacology to demonstrate that top-down cholinergic modulation of bulbar VPC activity via muscarinic receptors is a key player for social discrimination in rats. The manuscript is quite interesting, however there are some concerns with their results and how they are interpreted that need to be addressed.

In Figure 1, the authors show quantifications of the number of VPCs, pERK+ VPCs and pERK+ MCs. How was the quantification done? Some more details are needed in the methods paragraph in order for the reader to understand the stainings. Did the authors normalize the number of cells to the area? The authors should consider to present the normalized density of the cells between brain sections and animals. In addition, it is recommended to split the channels (green and magenda) in addition to showing the composite. This can help the reader visualize better the cells that are colocalized with GFP and pERK.

Reply: The counting was done in 4-6 non-overlapping sections of one olfactory bulb of a respective rat. The total number of positive cells was averaged over the number of counted sections, to account for variance in cell distribution and number of sections analysed. Double-labelled cells were counted, if the typical soma shape was visible without a staining of the nucleus area in the GFP-channel (green) and could be identified with the same outline in the pERK channel (magenta, see example in Fig. 1a). For control the localization of the nucleus was double-checked in the Dapi channel (blue).

Although the average section size was similar across groups, we realized that due to the spherical shape and the layer composition of the olfactory bulb the densities of VPCs and MCs negatively correlated with section size, resulting in an overestimation of cell densities in the smaller slices compared to bigger ones. Thus, a normalization of the number of VPCs and MCs does not seem appropriate for our analysis.

However, to follow up on the suggestion of the referee and to reduce variance, in our new Fig. 1 we now normalize the number of pERK+/VPCs per slice to the total number of VPCs per slice to account for a possible imbalance in section size or number of GFP-labelled VPCs (heterozygous expression of GFP) between the stimulation groups. We also added a detailed description of the quantification of pERK+/GFP cells in the methods section (ll. 679-691).

As suggested by the referee, we split the green and magenta channel in Fig 1a and Suppl. Fig 1 and added an insert panel with higher magnification of cells of interest to the Fig.5a. According to the suggestions of reviewer #1 and the editorial comments we also provide higher quality IF images (higher contrast by computational clearing and higher magnification).

In Figure 3A, the authors observed somatic inhibition upon ON stimulation. When they bath applied ACh, inhibition was reduced (Fig. 3B). Yet, they don't see Ca²⁺ signals being propagated to the soma (Fig. 3F). But, during ACh application, inhibition was reduced (A and B), so there should have been a change in somatic Ca²⁺ with calcium imaging. So, based on their findings of Fig. 3E-G, GABAergic inhibition does not account for the lack of somatic Ca²⁺ signaling, nor does ACh. In the text, the authors claim "The finding of a distal postsynaptic Ca²⁺ entry that cannot be detected near the soma, is consistent with the concurrent observation of strong GABAergic inhibition at the soma suggesting that excitation at the tufts is inhibited alongside the apical dendrite". This statement needs to be supported either experimentally or the authors should provide proper references.

Reply: We agree with the referee that the mentioned statement is not supported sufficiently by the experimental evidence, as we do not know the locus of either inhibition or ACh-mediated excitation on the VPC dendritic tree at this point. For this reason, we omitted the critical passage. This part reads now: "In conclusion, Ca²⁺ influx into the tuft or proximal apical dendrite appears to be neither required for, nor modified by the ACh-mediated modulatory effects on ON-evoked postsynaptic PSPs." (ll. 237-238)

In Figure 4, the authors performed immunostainings of the neurons in the diagonal band of Broca against pERK and choline acetyltransferase using the brains from the exposure experiments. They came to the conclusion that cholinergic cells in the HDB are activated during social interaction, however, they do not find any significant difference between the number of pERK+ cholinergic cells comparing between water and conspecific (Fig.4C). This should be explained in the text.

Reply: Indeed, there was no significant difference between water stimulation and social interaction in cholinergic cells of the HDB. As discussed in our manuscript, the HDB integrates various inputs like internal states, arousal, and sensation. Therefore, we cannot infer the reason for the relatively high activation during water presentation from our data. However, water presentation (with no or few olfactory cues) was not enough to increase pERK activity in VPCs to the same level as social interaction in our experiments, suggesting that both, ACh cell activation in the HDB and olfactory input are needed for VPC activation. In support, our in-vitro ephys results demonstrate that ACh modulation enables excitation in VPCs by olfactory nerve stimulation (mimicking odor stimulation). We added this suggestion to this issue in the discussion (ll. 480-488).

In Supplemental Figure 1, the authors should consider splitting the two channels (green and magenta). The colocalization of the cells is not evident. Also, a panel with higher magnification images could help the reader appreciate the work.

Reply: As suggested, we split the green and magenta channel in Fig 1a and Suppl. Fig 1 and added an insert panel with higher magnification of cells of interest to the Fig.5a. According to the suggestions of referee #1 and the editorial comments we also provide higher quality IF images (higher contrast by computational clearing and higher magnification).

Finally, in Supplemental Figure 2D the authors claim that electrical AOB stimulation does not evoke excitatory responses in MOB VPCs. The authors should show the representative averaged traces of responses instead of a single trace.

Reply: We changed the former representation in Supplemental Fig. 2d to a representative response of averaged traces.

Reviewer #3 (Remarks to the Author):

In this paper, the authors demonstrate the existence of a top-down regulation from the basal forebrain cholinergic system that, via VPC neurons, is involved in social discrimination. The authors show that activation of bulbar VPC neurons requires the additional activation of the basal forebrain cholinergic input acting via muscarinic and nicotinic receptors. While the experiments are generally straightforward, the conclusions are not. Overall, a lack of mechanistic explanations for the effects dampen my enthusiasm for the paper. Statistical approaches are adequate.

Major Concerns

1) If pERK is taken as a marker for neuronal activation, based on Fig. 1, it appears that the entire neuronal population of the bulb show activation, not just VPC neurons, upon social exposure. This suggests general arousal mechanisms rather than a specific modulation of VPC neurons. This might explain why lesser number of VPC neurons are activated with urine, that would only activate a subset of the glomerular circuits. Would the authors predict similar increase in VPC activation in all perceptual learning tasks, e.g., discrimination of closely related odors, that also involves the cholinergic inputs? The point here is that the results might be true for all behaviors involving attention and learning rather than a specific example of multisensory integration in the social paradigm used in this paper.

Reply: We want to thank the referee for this suggestion concerning the interpretation of our data. We now included the alternative hypothesis that cholinergic activation of VPCs leads to improved olfactory perceptual learning (including social discrimination) in our discussion. (ll. 544-553). We also emphasize now in the introduction that next to multisensory inputs also arousal and attention may be factors to be involved in modulating social discrimination. (ll. 80-82)

2) Bath application of agonists is not a good way to elicit responses. Firstly, it activates receptors across the bulb making it difficult to identify specific targets. Secondly, it is not an effective means for activating

receptors that can desensitize like the nicotinic receptors, which might explain partial effects seen with nicotinic blockade. In fact, the idea that nicotinic receptors might be involved in gain control mechanisms (J. Neurosci 32,3261,2012) similar to that discussed in the paper makes this an important point to address.

Reply: In our study, we did not try to identify specific targets of ACh release on VPC sub-compartments but were rather interested in the network effect of ACh administration or ACh release possibly via cholinergic projections in ON-evoked VPC signaling.

On point (1): VPC dendrites are located within the glomerular layer and shortly beneath it, indicating that these cells are unlikely to receive inputs from deeper layers, thus a selective application of agents to the glomerular layer would most likely not increase specificity and thus, bath application of ACh appeared as a legitimate choice for our experiment. We also wish to point out that acetylcholine release in the olfactory bulb following HDB stimulation has been detected by rather insensitive microdialysis (recovery rate of 14-16%, Uchida and Kagitani, J Physiol Sci 2018) and was found to last several minutes. Also, optogenetic stimulation of HDB cholinergic projections in the OB *in-vivo* increases odor responses in projection neurons even without specific odor stimulation (Rothermel et al., 2014, J Neurosci). This indicates that ACh is released rather widespread in the bulb during *in-vivo* condition.

On point (2): We cannot completely exclude the possibility of desensitization of nicotinic receptors, although we were not able to find changes in ACh modulated ON-evoked signals over the course of the bath application that indicates desensitization effects. Nevertheless, we now clearly state the spatial and temporal limitations of our application method in the discussion compare to local puff application (II. 441-446). In addition to the new experiments that we describe in the response to point number 3 of the referee, we now discuss the involvement of nicotinic receptor signaling in gain control mechanisms in the context of our findings following nicotine stimulation and nicotinic receptor blockade (II. 447-451).

Rothermel, M., Carey, R.M., Puche, A., Shipley, M.T., and Wachowiak, M. (2014). Cholinergic Inputs from Basal Forebrain Add an Excitatory Bias to Odor Coding in the Olfactory Bulb. *The Journal of Neuroscience* 34, 4654-4664.

Uchida, S., and Kagitani, F. (2018). Effect of basal forebrain stimulation on extracellular acetylcholine release and blood flow in the olfactory bulb. *The Journal of Physiological Sciences* 68, 415-423.

3) Coupled with the application method used, the pharmacology needs a more thorough elucidation. Mecamylamine at 20 μM is unlikely to block the $\alpha 7$ nicotinic receptors, which is densely expressed in the glomerulus. Similarly, atropine, at 10000-fold the K_i for muscarinic receptors (why use this high a concentration?) can also affect nicotinic currents. How did the authors conclude that the relevant receptors were completely blocked and that other types were not affected? At least two chemically distinct antagonists should be used to confirm receptor specificity. Also, the authors might use selective agonists (nicotine and muscarine, not oxotremorine that can have nicotinic effects).

Reply: We appreciate these comments by the referee, which substantially improved our insights into cholinergic pharmacology. We have now performed additional experiments investigating the effects of the selective agonists muscarine and nicotine on olfactory nerve evoked IPSPs in VPCs (II. 257-280, new Fig. 4). Although we cannot completely exclude that our mecamylamine concentration is too low to completely block all nicotinic receptors in the OB slice, our new results show that muscarine application consistently results in reduction of IPSP amplitudes or even AP or EPSP generation comparable to ACh application combined with mecamylamine. Moreover, although we cannot completely exclude that our

atropine concentration also blocks nicotinic receptors (concentrations were according to former literature, e.g. Pignatelli A, Belluzzi O. *Chemical Senses* (2008).), our new results show that nicotine application results in no statistical changes in ON-evoked IPSPs to IPSPs in ACSF. This is in line with the results from the atropine-ACh application. We changed the results section and the discussion according to our new results (see point 4). We further emphasize in the discussion that due to our experimental limitations we cannot fully exclude the involvement of nicotinic signaling in the modulation of ON-evoked signals in VPCs (see next point).

4) Based on the above two points, relative contributions of the two AChR subtypes cannot be estimated at this time.

Reply: According to the criticisms of the referee, we adapted the discussion according to our results including the new findings concerning nicotine and muscarine application (see point 3), now stating more carefully the relative contributions of AChR subtypes:

(1) "According to our data, mainly muscarinic but not nicotinic stimulation reduced ON-evoked IPSP amplitudes in VPCs." (ll. 437-438)

(2) Moreover, muscarine application did not entirely replicate ACh effects (Fig. 2g, Fig. 4f+h). Thus, we cannot fully exclude nicotinic action during VPC modulation. (ll. 450-451).

5) In the calcium imaging experiments, the authors should do the experiments where the calcium concentration is initially clamped at about 50 nM using a mixture of calcium chloride and BAPTA or EGTA. The concern is that certain components of calcium signaling, e.g., receptor-coupled ER store release might not be operational under the conditions used.

Reply: While we agree with the reviewer that the recording conditions are not ideally suited for a detailed investigation of Ca²⁺ signaling pathways, in the current study we used Ca²⁺ imaging mainly as a readout for local depolarization/location of excitatory inputs and were not primarily interested in local Ca²⁺ signaling as such. Moreover, with the very same internal solution (100 μM OGB-1), resting concentrations of ~ 50 nM were detected using the Maravall et al. (Biophysical J, 2000) method in granule cells, mitral cells and hippocampal pyramidal cells (Egger & Stroh J Physiol 2009, Rammes et al., Neuropharmacology 92, 2015), and release from ER stores was demonstrated to substantially contribute to intracellular postsynaptic Ca²⁺ signaling in granule cell spines (Egger et al., J Neurosci, 2005). Thus, we would assume that Ca²⁺ signaling would not be overly disrupted.

Further, we added additional analysis to support our justification: "Since 50 Hz-AP trains can elicit Ca²⁺ signals also close to the soma in the proximal apical dendrites, the lack of such signals in response to single APs in the presence of ACh is not a technical artefact but probably due to a very low density of voltage-gated Ca²⁺ channels in these compartments, as known from hypothalamic VPCs²⁸ (Supplemental Fig. 3b; ON/soma vs. 50Hz/soma p=0.014; n=8 from 8 rats). Moreover, in the imaging experiments (with 100 μM OGB-1 in the internal solution) there was no difference in the effect of ACh on the increase of evoked PSPs compared to the previous electrophysiological experiments (Supplemental Fig. 3c, p=0.456; n=9 vs. 11), and therefore this effect is independent of the potential buffering of postsynaptic Ca²⁺ by the Ca²⁺ indicator. In conclusion, Ca²⁺ influx into the tuft or proximal apical dendrite appears to be neither required for, nor modified by the ACh-mediated modulatory effects on ON-evoked postsynaptic PSPs." (ll. 227-238)

REVIEWERS' COMMENTS:

Reviewer #1 (Remarks to the Author):

The manuscript generally improved and only a few points remain open:

- 1) The title "Top-down acetylcholine contributes to social discrimination via enabling action potentials in olfactory bulb vasopressin cells" is still misleading. No direct causal link from "Top-down acetylcholine" to "social discrimination" or even "action potentials generation in olfactory bulb vasopressin cells" has been demonstrated. This needs to be modified.
- 2) It seems that Supp. Fig 4 is only mentioned in the Methods. These findings should be mentioned in the results and the figure (or at least part of it) should also be integrated into a figure of the main MS.
- 3) The same is true for the control experiments in Supp Fig.3
- 4) Fig 1e: the asterisk above the rat group is misleading since two groups are significant: Line 121 (Fig. 1b+e; $p=0.042$, ANOVA, rat vs. water $p=0.025$, rat vs. urine $p=0.034$, water vs. 122 urine $p=0.949$)

Reviewer #2 (Remarks to the Author):

The authors have now fulfilled the requested changes and I approve the paper for publication.

Reviewer #3 (Remarks to the Author):

My main concerns in the original manuscript were overly strong conclusions not warranted by the results. The authors have reworked the manuscript and the current modified version is more circumspect in data interpretation. I have no additional concerns.

Reviewer #1 (Remarks to the Author):

The manuscript generally improved and only a few points remain open:

1) The title “Top-down acetylcholine contributes to social discrimination via enabling action potentials in olfactory bulb vasopressin cells” is still misleading. No direct causal link from “Top-down acetylcholine” to “social discrimination” or even “action potentials generation in olfactory bulb vasopressin cells” has been demonstrated. This needs to be modified.

Reply: We changed the title to the suggestion provided by the editor: “Top-down acetylcholine signaling via olfactory bulb vasopressin cells contributes to social discrimination in rats”

2) It seems that Supp. Fig 4 is only mentioned in the Methods. These findings should be mentioned in the results and the figure (or at least part of it) should also be integrated into a figure of the main MS.

Reply: According to the suggestion of the reviewer, we now mention these behavioral controls in the results section (p. 11-12). Further, we integrated the former supplementary figures (S4c+d) concerning the play behavior and the habituation towards a juvenile into the main Fig. 6.

3) The same is true for the control experiments in Supp Fig.3.

Reply: According to the suggestion of the referee, we now integrated the former supplementary figures (S3a+b) concerning the 50 Hz somatic stimulation control experiment into the main Fig. 3.

4) Fig 1e: the asterisk above the rat group is misleading since two groups are significant: Line 121 (Fig. 1b+e; $p=0.042$, ANOVA, rat vs. water $p=0.025$, rat vs. urine $p=0.034$, water vs. 122 urine $p=0.949$).

Reply: We now indicate the significant difference between the rat group and both other groups (Fig. 1e).

Reviewer #2 (Remarks to the Author):

The authors have now fulfilled the requested changes and I approve the paper for publication.

Reply: Thanks.

Reviewer #3 (Remarks to the Author):

My main concerns in the original manuscript were overly strong conclusions not warranted by the results. The authors have reworked the manuscript and the current modified version is more circumspect in data interpretation. I have no additional concerns.

Reply: Thanks.